# Carbon sequestration by multiple biological pump pathways in a coastal upwelling biome

**Michael R. Stukel** [1,2] ✉, **John P. Irving**[1,2], **Thomas B. Kelly** [1,3], **Mark D. Ohman**[4], **Christian K. Fender**[1] **& Natalia Yingling**[1]

Multiple processes transport carbon into the deep ocean as part of the biological carbon pump, leading to long-term carbon sequestration. However, our ability to predict future changes in these processes is hampered by the absence of studies that have simultaneously quantified all carbon pump pathways. Here, we quantify carbon export and sequestration in the California Current Ecosystem resulting from (1) sinking particles, (2) active transport by diel vertical migration, and (3) the physical pump (subduction + vertical mixing of particles). We find that sinking particles are the most important and export 9.0 mmol C m$^{-2}$ d$^{-1}$ across 100-m depth while sequestering 3.9 Pg C. The physical pump exports more carbon from the shallow ocean than active transport (3.8 vs. 2.9 mmol C m$^{-2}$ d$^{-1}$), although active transport sequesters more carbon (1.0 vs. 0.8 Pg C) because of deeper remineralization depths. We discuss the implications of these results for understanding biological carbon pump responses to climate change.

Photosynthesis by phytoplankton in the sunlit surface ocean decreases the partial pressure of dissolved $CO_2$, leading to the net uptake of $CO_2$ from the atmosphere. However, most phytoplankton carbon will be rapidly respired back into the surface ocean (period of days to a week). Long-term carbon sequestration requires transport of the organic matter fixed by phytoplankton into the deep ocean via a suite of processes collectively referred to as the biological carbon pump (BCP)[1,2]. The BCP likely transports 5–12 Pg C yr$^{-1}$ into the deep ocean, and even small changes in the BCP can have substantial effects on atmospheric $CO_2$ levels[3–6]. However, the response of the BCP to future climate change is uncertain[7].

While early BCP research focused on sinking particles, recent evidence highlights the importance of a suite of different processes: vertically migrating organisms consume organic matter in the euphotic zone and migrate to the deep ocean, where they can respire this organic matter and/or die in the mesopelagic[8–10]. Physical processes also directly inject particles and dissolved organic matter into the deep ocean as part of the eddy subduction pump[11,12] or through vertical mixing[13–15]. These distinct mechanisms of carbon transport likely lead to different depths of remineralization and commensurately

different sequestration durations[2]. They will also respond differently to expected climate change impacts on the oceans, including increased stratification in the open ocean, elevated upwelling rates in eastern boundary currents[16,17], and direct impacts of warmer temperatures and ocean acidification. However, due to the disparate spatiotemporal scales over which these processes act and the difficulty of making in situ biogeochemical rate measurements, these processes have not been simultaneously quantified from field measurements in any marine ecosystem.

Here, we compile data from 11 (~month-long) cruises conducted in the California Current Ecosystem (CCE) by the CCE Long-Term Ecological Research (LTER) program to simultaneously quantify organic carbon flux mediated by sinking particles, vertical migrants, and physical transport of particles. The CCE study region is an eastern boundary current upwelling biome that spans a productivity gradient including: (1) a high-nutrient, high-biomass coastal upwelling zone[18,19], (2) an often-Fe-limited mesotrophic region supported by cross-shore fluxes from the coastal area and diffuse wind-stress curl upwelling[20–22], and (3) an oligotrophic offshore region that is characteristic of the broad North Pacific Subtropical Gyre[18,23]. It thus spans much of the

[1]Earth, Ocean, and Atmospheric Science Dept., Florida State University, Tallahassee, FL, USA. [2]Center for Ocean Atmosphere Prediction Studies, Florida State University, Tallahassee, FL, USA. [3]College of Fisheries and Ocean Sciences, University of Alaska Fairbanks, Fairbanks, AK, USA. [4]Scripps Institution of Oceanography, University of California-San Diego, La Jolla, CA, USA. ✉e-mail: mstukel@fsu.edu

productivity variability found in the global ocean. In this study, we combine extensive in situ measurements spanning nutrients and nonliving organic matter to zooplankton with synthetic data-assimilative modeling approaches to quantify the export pathways of the BCP, their characteristic remineralization length scales, and the amount of $CO_2$ sequestered in the CCE via each process.

## Results and discussion
### Sinking particle flux
We quantified organic carbon flux associated with sinking particles using free-drifting sediment traps ($n = 99$ triplicate measurements). Export across the 100-m depth horizon was highly variable across the region (range = 2.6–41 mmol C m$^{-2}$ d$^{-1}$, Fig. 1d, Supplementary Data 1), as expected from the dynamic ecological landscape (Fig. 1a–c). Much of this variability was driven by cross-shore gradients in net primary production (NPP, Fig. 1a), although export did not decrease monotonically with distance from shore due to the complex mosaic of intersecting ecological communities created by mesoscale dynamics and the temporal lags generated as organic carbon moved through the

ecosystem[20,24]. By combining satellite-observed NPP with an empirical fit between sinking particle flux and contemporaneous NPP measured in situ[25], we determined that average regional carbon export via sinking particles was $9.0 \pm 2.2$ mmol C m$^{-2}$ d$^{-1}$. For comparison, global satellite-derived estimates of sinking particle flux predict ocean-average values of ~3 mmol C m$^{-2}$ d$^{-1}$ at the 100-m depth horizon[5] and 3–8 mmol C m$^{-2}$ d$^{-1}$ at the base of the euphotic zone[3,4,26,27]. Thus, the CCE is a site of above-average particle flux, although not substantially outside typical ocean values. Considering spatial variability, the heterogeneous CCE region includes offshore areas that fall slightly below the expected ocean average, as well as a coastal upwelling domain where sinking particle flux is substantially higher than the global average.

Using a Bayesian statistical framework, we determined that organic carbon flux decreased with depth following a power-law relationship with an average exponent of $b = 0.72$ (95% CI = 0.68–0.76). This exponent is on the lower end of values estimated globally ($b$ estimated to vary from ~0.4 to 1.6[28]). Relatively weak flux attenuation with depth is not surprising for the CCE, where diatoms (with silica

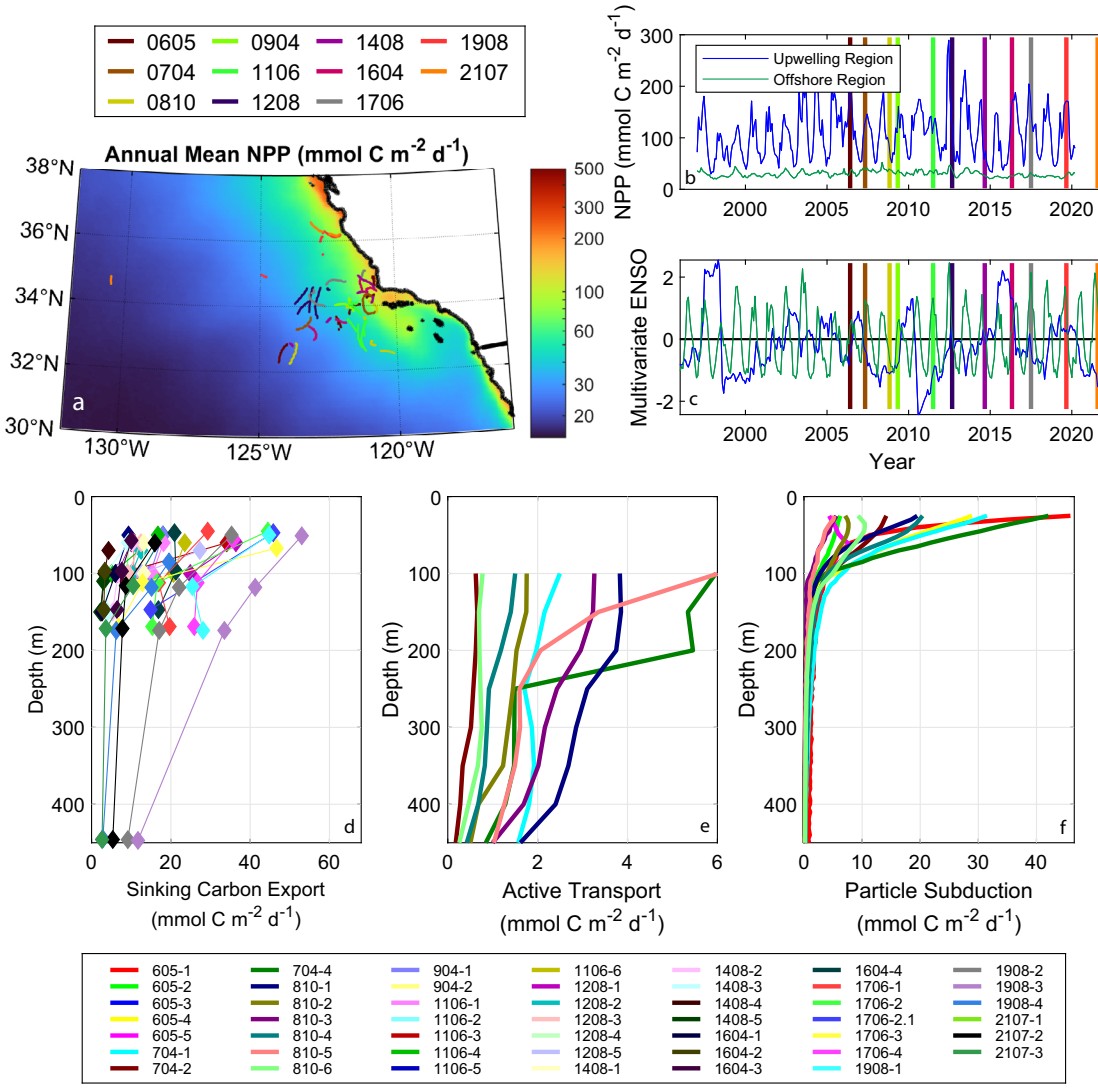

**Fig. 1 | Spatiotemporal variability in California Current Ecosystem properties and carbon export. a** Spatial map of mean net primary production (NPP)[64]. **b** Time-series of net primary productivity near the Point Conception upwelling center and in the offshore, oligotrophic region. **c** Time-series of the multivariate El Niño-Southern Oscillation index and normalized Bakun upwelling index. **d–f** Carbon export via **d** sinking particles, **e** active transport mediated by diel vertically

migrating zooplankton, and **f** particle subduction. In (**a–c**), colors are indicative of cruise. In (**d–f**), colors indicate results from different Lagrangian experiments (labeled as 'Cruisename-experiment#'; not all experiments had data available for all biological carbon pump (BCP) pathways). Note the different *x*-axis scales. Source data are provided as a Source Data file.

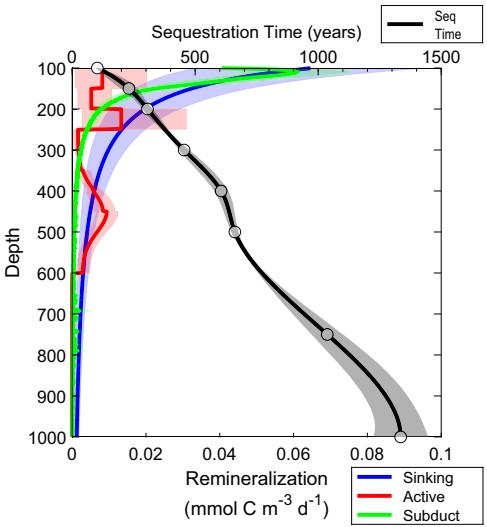

**Fig. 2 | Regional average remineralization as a function of depth for the three carbon export pathways (blue = sinking, red = active transport, green = subduction and vertical mixing; bottom axis).** Top axis and black line show sequestration duration as a function of remineralization depth. Shaded colors are 95% CI. Source data are provided as a Source Data file.

frustules) are important contributors to the phytoplankton community, rapidly sinking fecal pellets are dominant contributors to sinking particle flux, temperatures are modest (5.3–6.5 °C at 500 m depth), and low subsurface oxygen concentrations (11–30 µmol kg$^{-1}$ at 500 m) may limit remineralization[29–31]. The CCE *b* value equates to a 39% decrease in flux from 100 to 200 m depth. We estimate that average flux declined to 5.5 ± 1.3, 2.8 ± 0.7, and 1.7 ± 0.4 mmol C m$^{-2}$ d$^{-1}$ at the 200 m, 500 m, and 1000 m depth horizons, respectively.

### Active transport by diel vertical migrants
The biomasses and daytime residence depths of diel vertically migrating (DVM) zooplankton taxa (copepods, krill, chaetognaths, and hyperiid amphipods) were determined from day-night-paired, vertically stratified tows from the surface to 450 m[30]. DVM mesopelagic fish biomass was quantified using Matsuda-Oozeki-Hu net trawls[32]. Bioenergetics models and allometric relationships were then used to quantify the respiration and excretion of these taxa at their daytime residence depths. Zooplankton active transport across 100-m depth ranged from 0.3 to 5.1 mmol C m$^{-2}$ d$^{-1}$ (mean = 2.1, 95% CI of the mean = 1.1–3.3) and was higher in productive coastal regions than offshore areas. Fish active transport was typically a factor of 2 lower (range = 0.25–1.4, mean = 0.8, CI = 0.6–1.1). Summed active transport averaged 2.9 mmol C m$^{-2}$ d$^{-1}$ (CI = 1.9–4.1) at 100 m and declined gradually with depth during most Lagrangian experiments (Fig. 1e). However, during two Lagrangian experiments (both krill-dominated) the daytime residence depth of most DVM zooplankton was <250 m, leading to more rapid decreases in flux with depth. Regional average respiration rates of DVM taxa peaked in the 200–250 m depth range at 0.013 mmol C m$^{-3}$ d$^{-1}$, although these estimates had substantial uncertainty (95% CI = 0.003–0.031) and were not statistically different from respiration rates from 100 to 200 m depth (Fig. 2). Respiration was comparatively low from 250 to 350 m but reached another peak near 450 m depth (0.009 mmol C m$^{-3}$ d$^{-1}$, 0.006–0.012), where mesopelagic fish typically reside[33].

These estimates of excretion and respiration active transport and daytime residence depth are similar to results from an independent global model[34] and a global synthesis of acoustic data[35], respectively. These studies further highlight the fact that the CCE has comparatively high active transport relative to other ocean regions and

comparatively shallow zooplankton migration depths. Our estimates are, however, likely underestimates of the total magnitude of active transport in the region because biomass can be underestimated when actively swimming taxa avoid nets and because our estimates do not include other processes that contribute to active transport, including foraging forays[36] and migrant mortality at depth. Mortality at depth, in particular, may be responsible for approximately half of the total active transport in the region[37]. Mortality at depth is also likely to lead to deeper remineralization due to the sinking of carcasses and/or fecal pellets of predators that feed on vertical migrants[38]. However, mortality rates in the mesopelagic are highly uncertain due to a paucity of direct measurements, and hence we cannot robustly include this process in our carbon budget.

### Subduction and vertical mixing
We estimated particulate organic carbon subduction and vertical mixing rates by combining sinking speed spectra of rapidly and slowly sinking particles estimated from our field data with a Lagrangian particle tracking model coupled to circulation fields from dynamically consistent data-assimilating physical state estimates simulating three-dimensional currents during our cruises[39,40]. Subduction from the euphotic zone was substantial (Fig. 1f, Supplementary Data 2), but flux attenuation was rapid. By 100-m depth, flux had declined to a regional average of 3.8 mmol C m$^{-2}$ d$^{-1}$ (95% CI = 3.0–4.6), and 74% of this particulate organic carbon was remineralized before reaching 200 m depth (95% CI = 65–80%). Subduction and vertical mixing only transported 0.20 (0.13–0.29) mmol C m$^{-2}$ d$^{-1}$ across the 500-m depth horizon and 0.059 (0.035–0.087) mmol C m$^{-2}$ d$^{-1}$ across the 1000 m depth horizon. The declining importance of subduction with depth is found both when considering transfer efficiency (T$_{100}$ = export at a depth of 100 m deeper than the euphotic zone/export at the base of the euphotic zone, Fig. 3a), which was typically -0.2, and when evaluating the relative contributions of subduction to total export flux across different depth horizons (Fig. 3b). Subduction and vertical mixing were often important and occasionally even dominant components of export flux across the 100-m depth horizon but never contributed >20% of export at 200- or 300-m depth. Thus, while the physical pump was important to the euphotic zone carbon budget, it contributed little to export through the mesopelagic. These estimates do not include the contribution of dissolved organic molecules to carbon export, which has been shown to be important in some regions that experience deep winter mixed layers[13]. Our estimates should therefore be considered conservative, although since the CCE does not experience deep convective winter mixing, we believe that the general patterns and magnitude of flux are robust.

### Sequestration
To quantify CO$_2$ sequestration time as a function of depth, we ran a 500-year Lagrangian simulation of the North Pacific and released simulated CO$_2$ at depths from 100 to 1000 m in the CCE. Although most CO$_2$ molecules respired at 100 m depth were entrained into the mixed layer within the first few years of the simulation, mean sequestration time was substantially longer (102 ± 3 years) because a substantial portion of the molecules was advected westward into the subtropical gyre and remained sequestered for centuries. Mean sequestration time increased approximately linearly with depth and reached 1335 ± 57 years at a depth of 1000 m (Fig. 2, Supplementary Data 3).

We quantified the amount of carbon sequestered by each BCP pathway as the integral over depth of the product of sequestration time and remineralization rate (Table 1), where remineralization rate refers to respiration + excretion for active transport and to the derivative of carbon flux with depth for sinking and subducted particles. Sequestration was substantially greater for sinking particles (1931 mol C m$^{-2}$) than active transport (498 mol C m$^{-2}$) or subduction

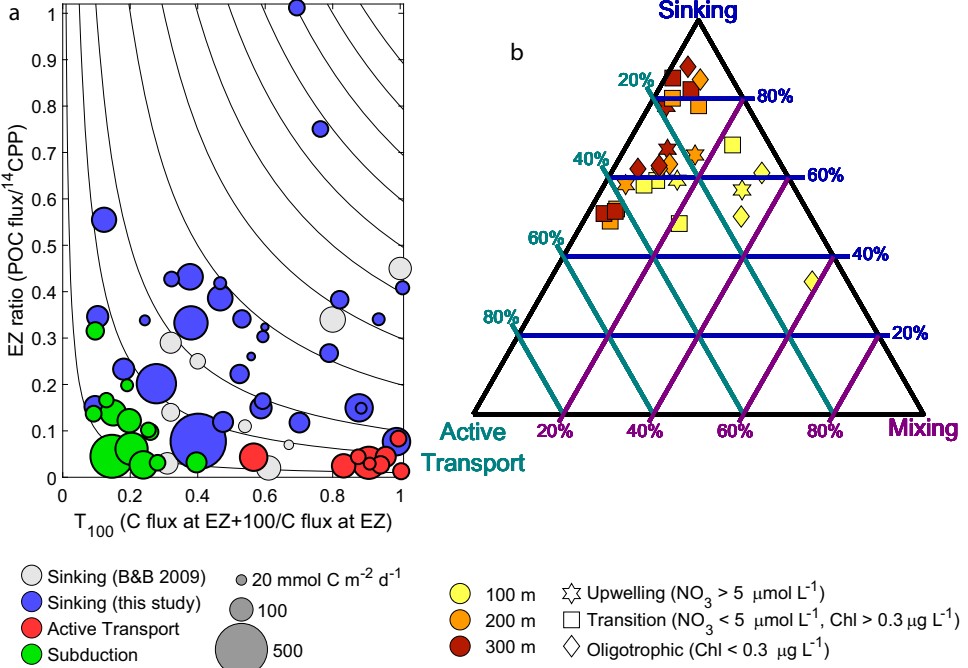

**Fig. 3 | Varying importance of biological carbon pump pathways with depth for individual Lagrangian experiments. a** EZ ratio (export efficiency at the 0.1% light level) vs. $T_{100}$ (transfer efficiency through the mesopelagic) following ref. 79. Gray symbols show results of sinking flux in other regions as compiled by ref. 79. Blue, red, and green are sinking, active transport, and subduction, respectively, in the California Current Ecosystem. Symbol size is proportional to net primary production.

**b** Triangle diagram showing the proportion of export due to each biological carbon pump (BCP) pathway at different depth horizons following ref. 80. Symbols near the top vertex indicate sinking dominated export; symbols near the bottom left are dominated by active transport; bottom right are mixing. Note that panel (**b**) shows data only from Lagrangian experiments with simultaneous measurements of all three pathways. Source data are provided as a Source Data file.

**Table 1 | Regionally averaged (and 95% confidence limits) contributions of each biological carbon pump pathway**

|  | Sinking | Active transport | Subduction |
|---|---|---|---|
| **Export at 100 m (mmol C m⁻²)** | 9.0 (6.8–11.2) | 2.9 (1.7–4.2) | 3.8 (3.0–4.6) |
| **Mean sequestration duration (y)** | 586 (554–621) | 468 (401–559) | 279 (248–314) |
| **Regional sequestration (mol C m⁻²)** | 1931 (992–2848) | 498 (327–669) | 386 (308–467) |
| **Regional sequestration (Pg C)** | 3.9 (2.0–5.8) | 1.0 (0.7–1.4) | 0.8 (0.6–1.0) |

(386 mol C m⁻²). The higher sequestration due to sinking particles was a result of both a higher flux from the surface ocean (9.0 mmol C m⁻² d⁻¹ at 100 m depth, compared to 2.9 and 3.8 for active transport and subduction, respectively) and a longer mean sequestration time (586 years, compared to 468 and 279). Integrated over the $1.7 \times 10^5$ km² southern CCE study region (see "Methods"), the BCP sequesters a total of 5.7 Pg C (Fig. 4). For comparison, in situ measurements[41] show that this region contains 13.6 Pg C of inorganic carbon, which suggests that the BCP is responsible for approximately half of the carbon sequestered in the CCE (with the solubility pump responsible for the remainder).

**Spatiotemporal variability in BCP pathways in the CCE**
Our study region is spatiotemporally variable and includes both a coastal upwelling domain and an oligotrophic offshore region that is contiguous with and biogeochemically similar to the North Pacific subtropical gyre. To understand how BCP pathways vary within the mosaic of communities that exist within the CCE, we computed Spearman's rank correlations between the proportion of NPP that was exported by each pathway and important ecosystem metrics, including surface temperature, surface nitrate, surface chlorophyll, and vertically integrated NPP (Supplementary Table S3). Notably, the

proportion of NPP exported by subduction and sinking particle flux showed positive correlations with temperature and negative correlations with surface nitrate, surface chlorophyll, and vertically integrated NPP (all statistically significant except subduction vs. temperature). The proportion of NPP exported by active transport was not significantly correlated with any ecosystem metrics. This suggests that active transport is relatively more important to export during upwelling conditions. These results also highlight the fact that total export efficiency (i.e., the sum of all BCP pathways/NPP) is inversely related to local ecosystem productivity. This inverse relationship has been previously noted for the CCE with respect to sinking particle flux[25,42] and is likely driven by spatiotemporal decoupling of production and export[20,24,43].

This connectivity between coastal and offshore regions, driven by a combination of large-scale Ekman transport and smaller-scale meso- and submesoscale features, complicates simple interpretations of temporal variability in BCP pathways. For instance, while spatial correlations show an inverse relationship between upwelling and the efficiency of export via sinking particles and subduction, upwelling and commensurate offshore Ekman transport of organic matter are likely causes of the high export efficiency found offshore. Thus, periods of upwelling likely increase the strength of the subduction pump

because coastal and/or wind-stress curl upwelling must be paired with downwelling/subduction in another region. Simultaneously, upwelled nutrients stimulate the growth of diatoms which provide mineral ballast for sinking particles and promote shorter food-web pathways to mesozooplankton that produce most of the recognizable sinking material in the region[44,45] and upwelling-favorable conditions (e.g., La Niña) favor the dominant vertically migrating zooplankton[46]. It is thus reasonable to surmise that coastal upwelling promotes all three BCP pathways, although it may have the greatest local impact on active transport in the coastal subdomain while also substantially enhancing sinking particle flux throughout the broader domain. Importantly, coastal upwelling shows predictable patterns in the CCE across multiple temporal scales. Seasonally, coastal upwelling peaks in spring, while on interannual time scales, it is greater during La Niña conditions than during El Niños[47]. It is also increasing over long time periods as a result of climate change and concomitant increases in land-sea temperature differences[48], suggesting long-term alterations in the BCP in the CCE.

### An interconnected BCP in a changing ocean

Climate change is impacting the ocean in myriad ways that affect the BCP. Surface heating leads to increased stratification in open-ocean regions, while wind intensification may increase coastal upwelling[17,48,49]. Slowing of the Atlantic Meridional Overturning

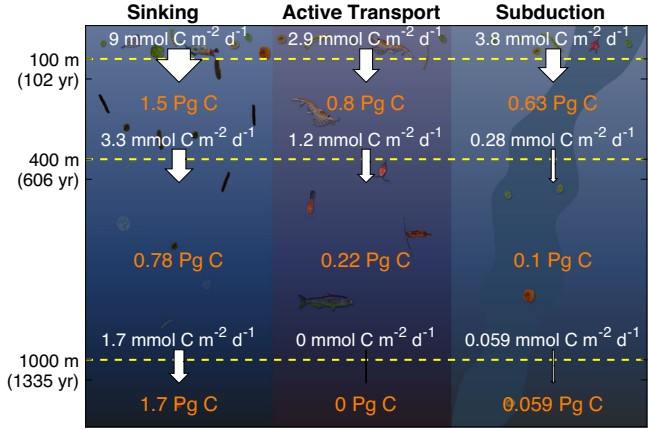

**Fig. 4 | Contributions of sinking particles, active transport, and subduction/vertical mixing to export (white arrows and white text) and carbon sequestration (orange text) in three depth horizons (100–400 m, 400–1000 m, and >1000 m) in the California Current Ecosystem.** Active transport does not include mortality at depth or fecal pellet production beneath the euphotic zone.

Circulation is decreasing ocean interior ventilation rates[50]. Increased temperatures are having direct impacts on multiple species and on oxygen saturation[49]. Ocean acidification is influencing multiple trophic levels and most directly affecting calcifying organisms that contribute disproportionately to sinking flux while contributing to the carbonate counterpump[51]. Increased $CO_2$ may also have stimulatory effects on some phytoplankton taxa[52]. Notably, these drivers of BCP change will alter each BCP pathway in distinct ways. Some (e.g., changes in primary productivity) are likely to have similar directional changes for all BCP pathways. However, others will lead to distinctly different effects. For example, the spread of oxygen minimum zones will likely decrease microbial respiration associated with sinking particles (leading to deeper remineralization of sinking particles) while forcing a shoaling of the daytime resident depth of migrating organisms[53]. This latter effect may already be leading to shallower respiration depths in the CCE[33] and contributing to shorter sequestration time scales for active transport relative to sinking particles (Table 1). Increased stratification will likely decrease all BCP pathways but affect subduction and vertical mixing rates most strongly. Conversely, increased coastal upwelling will likely substantially strengthen sinking and active transport pathways while having a muted effect on subduction rates because the downwelling associated with Ekman transport is likely to occur offshore in more oligotrophic regions relative to the coastal zones that will see enhanced productivity. Changing ocean ventilation rates will disproportionately impact different pathways insofar as they differentially impact sequestration temporal horizons with depth.

Addressing the overall impact of climate change on the BCP—and hence determining whether the BCP will be a positive or negative feedback on climate change—thus requires careful budgets of the relative importance of different pathways[7]. Our study enables such research by presenting the most complete observational BCP budget in the world ocean and shows that sinking particles are the most important pathway in the CCE. However, active transport and subduction/vertical mixing are both responsible for meaningful export (2.9 and 3.8 mmol C m⁻² d⁻¹, respectively) and sequestration (1.0 and 0.8 Pg C, respectively) and are likely underestimated due to our inability to constrain migrant mortality at depth and vertical transport of dissolved organic molecules. Our results shed light on carbon sequestration within eastern boundary current upwelling regions that are responsible for 2–5% of marine primary productivity but contribute disproportionately to fish production[54,55]. Increasing coastal upwelling will likely lead to increased production in these regions while also driving a shift toward even greater importance of sinking particles (which are already responsible for 57% of export in the CCE, Fig. 1) relative to other BCP pathways. The longer sequestration times of sinking particles (Table 1), combined with an expected increase in the remineralization length scale for sinking particles as the oxygen

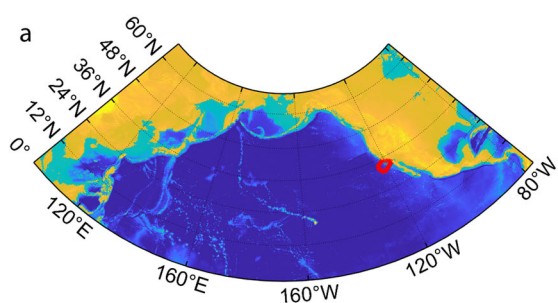

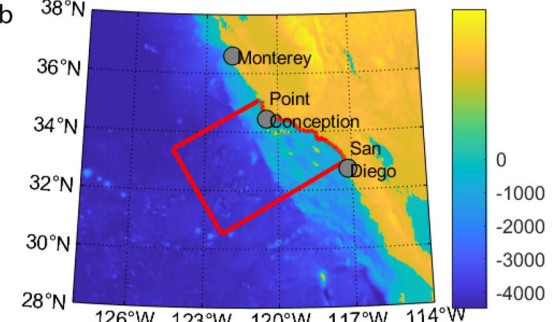

**Fig. 5 | Study domain. a** Model domain used to determine sequestration time as a function of depth. **b** Study region. The red box in each is the control volume used for integrations. Color is bathymetry from NOAA ETOPO1 (NOAA Technical Memorandum: Amante, C. and B.W. Eakins, 2009. ETOPO1 1 Arc-Minute Global Relief

Model: Procedures, Data Sources and Analysis. NOAA Technical Memorandum NESDIS NGDC-24. National Geophysical Data Center, NOAA. https://doi.org/10.7289/V5C8276M).

minimum zone shoals, thus suggest that the BCP in eastern boundary currents will likely act as negative feedback on climate change.

We should anticipate, however, that BCP changes will differ substantially in other biomes. For instance, oligotrophic subtropical gyres may be even more heavily weighted toward sinking particles than the CCE, although vertical migrators in oligotrophic regions are likely to reside at even deeper daytime depths[10,35]. In areas with deep winter mixed layers and/or substantially weaker stratification year-round (e.g., the North Atlantic and Southern Ocean), we might expect that subduction/vertical mixing contributes more to export from the euphotic zone while also having a substantially longer remineralization length scale[13,15]. Polar and subpolar regions also have ontogenetic (seasonal) vertical migrators that contribute to the "lipid pump[9]". Additionally, sequestration times are spatially variable, with the North Pacific exhibiting some of the longest sequestration periods while the North Atlantic and Southern Ocean have substantially shorter time scales[56–58].

We must also consider the interconnectedness of the BCP pathways. While our conceptual model differentiates three processes, it necessarily simplifies the complexity of the BCP. For instance, vertical migrators with long gut turnover times will defecate a portion of their ingested prey at depth (active transport), where it will be released as fecal pellets that contribute to further downward sinking flux. Conversely, sinking particles can be consumed by vertical migrators that transport the carbon deeper into the ocean. Organic molecules cleaved from sinking particles by microbial hydrolytic exoenzymes[59] may be subducted further into the ocean interior, while subducted particles can aggregate in the deep ocean to form sinking marine snow[60]. Slowly sinking particles in regions of active subduction or mixed layer deepening contribute simultaneously to multiple BCP pathways[39]. Distinctly different physical processes that we have lumped into the "subduction/vertical mixing" pathways (e.g., eddy pump, diapycnal diffusion, large-scale downwelling, mixed layer entrainment/detrainment[2,14]) may also respond differently to climate change while driving regionally variable remineralization length scales. By quantifying for the first time biogeochemical processes from production to sequestration across multiple BCP pathways, the present study provides an integrated framework to evaluate other biomes, approaches, and hypotheses.

## Methods
### Field studies
Samples were collected during 11 (typically month-long) process cruises of the California Current Ecosystem (CCE) Long-Term Ecological Research (LTER) program from 2006 to 2021. These cruises sampled the substantial spatiotemporal variability existing in the region while targeting a range of seasons (although most were during spring or summer and none were in winter) and occurring during different phases of the El Niño-Southern Oscillation. Cruise plans were designed around quasi-Lagrangian experiments during which in situ arrays with satellite-enabled surface drifters and subsurface 3-m long × 1-m in diameter holey-sock drogues were used to track mixed layer water parcels for a typical duration of ~4 days[61]. These quasi-Lagrangian experiments allowed extensive, repeated sampling of ecological and biogeochemical rates and standing stocks, enabling the development of detailed biogeochemical budgets.

### Net primary production
NPP was measured by the $H^{14}CO_3^-$ uptake method daily at 6–8 depths spanning the euphotic zone during each Lagrangian experiment[45]. Triplicate 250-ml samples (plus additional "dark" controls) were spiked with $H^{14}CO_3^-$ and placed in coarse mesh bags, which were subsequently attached to an in situ incubation array[61] at the depth of collection and incubated in the water column for 24 h. Samples were then recovered

and filtered onto GF/F filters, and $^{14}C$ activity was determined using a liquid scintillation counter.

### Sinking particles
Sinking particles were collected using VERTEX-style free-drifting sediment traps[62]. The sediment trap array included surface floats, a subsurface holey-sock drogue centered at a depth of 15 m in the mixed layer, and crosspieces holding typically 8–12 particle interceptor tubes (70-mm inner diameter, 8:1 aspect ratio, topped with a baffle comprised of 13 smaller tubes that were tapered at the top). The placement (and number) of crosspieces varied but typically included one near the base of the euphotic zone (defined using the 0.1% surface irradiance criterion) and one (or more) crosspieces at deeper depths. Tubes were deployed with a formaldehyde brine. After recovery, metazoan "swimmers" were carefully removed from the filter under 20X magnification on a stereomicroscope, and samples were filtered through pre-combusted GF/F filters. Samples from three tubes were acidified with fuming HCl to remove inorganic C, then analyzed using either an elemental analyzer (for C/N analyses) or an isotope ratio mass spectrometry coupled to an elemental analyzer (for C/N and isotopes). Simultaneous measurement of $^{234}Th$ flux into the trap tubes and $^{234}Th$ flux quantified using the $^{238}U$–$^{234}Th$ approach have shown that our methodology has no substantial over- or under-collection bias[63].

To determine a regional, annual estimate of sinking particle flux, we previously developed a relationship between in situ sinking carbon flux, NPP, and temperature[25]: Sinking particle flux = e-ratio × NPP, where e-ratio = 0.056 (±0.008) × SST − 0.698 (±0.122), $R^2 = 0.67$, $p < 0.001$. We then applied this relationship to an NPP dataset derived from a multi-satellite merged algorithm optimized for the CCE region[64] to quantify spatiotemporal variability in sinking particle flux and determine regional mean sinking particle flux ($\bar{F}_{100}$) at the 100 m depth horizon[25].

To quantify the attenuation of particle flux with depth, we used a Bayesian statistical framework. We assumed a power law functional form for the decrease in particle flux with depth[65]. We then quantified the sum of squared normalized misfits (SSNM) between all sediment trap flux measurements and a power law estimate of export flux:

$$SSNM = \sum_{i=1}^{n} \sum_{j=1}^{n_i} \frac{\left( F_{ST,i,j} - f_{i,z=100} \left( z_{i,j}/100 \right)^{-b} \right)^2}{\sigma_{F_{ST,i,j}}^2} \quad (1)$$

where $n$ is the number of sediment trap deployments, $n_i$ is the number of depths at which sediment trap crosspieces were placed during deployment $i$, $F_{ST,i,j}$ is the sediment-trap-derived carbon export measurement at depth $j$ during deployment $i$, $f_{i,z=100}$ is the actual (and unknown) carbon export at 100 m depth during deployment $i$, $z_{i,j}$ is the depth of crosspiece $j$ during deployment $i$, $b$ is the average exponent defining carbon attenuation in the CCE, and $\sigma_{F_{ST,i,j}}$ is uncertainty in the sediment trap flux measurement for deployment $i$ and depth $j$ (which we assume to be the standard deviation of triplicate measurements). We then used a Markov Chain Monte Carlo (MCMC) procedure[66] to estimate $b$ (as well as $f_{i,z=100}$ for each deployment). Specifically, we started with initial guesses for $b$ ($b_0 = 1.2$ from ref. [28]) and for each $f_{i,z=100}$ ($f_{i,z=100,0} = 9$ mmol C m$^{-2}$ d$^{-1}$, which is the regional average computed above). We then proposed a new value for each parameter ($b$ and each $f_{i,z=100}$) by drawing a random number from a normal distribution centered at the previous value for each parameter. We then re-computed SSNM based on the new values and accepted these new values with probability:

$$prob = \frac{e^{-1/2 \times SSNM_{n+1}}}{e^{-1/2 \times SSNM_n}} \times \frac{prior_{n+1}}{prior_n} \quad (2)$$

where $prior_n$ and $prior_{n+1}$ represent the prior density of the parameters evaluated at the location of the previous parameter value and at the location of the proposed parameter value, respectively. For $b$, we used a normal prior distribution with a mean of 1.2 and a standard deviation of 0.5 based on ref. 28. For all $f_{i,z=100}$, we assumed a uniform prior distribution from 0 to 5000 (i.e., we assumed no knowledge of export flux at the 100 m depth horizon prior to measurement except that export flux had to have been between 0 and 5000 mmol C $m^{-2}$ $d^{-1}$). We ran the MCMC algorithm for a total of $10^7$ iterations and discarded the initial $10^6$ iterations as a "burn-in" period. This Bayesian approach provides a robust estimate of the regional mean value for the exponent ($\bar{b}$) of the power law attenuation equation along with confidence limits. We then define regional sinking carbon flux as a function of depth (at depths deeper than 100 m) using the equation:

$$\bar{F}_z = \bar{F}_{100} \times \left(\frac{z}{100}\right)^{-\bar{b}} \qquad (3)$$

Uncertainty in $\bar{F}_z$ was determined using a nonparametric Monte Carlo simulation to propagate uncertainty in $\bar{F}_{100}$ and $\bar{b}$.

### Active transport by diel-vertically migrating (DVM) zooplankton and fish

We estimated carbon transport to depth via the respiration and excretion of DVM zooplankton from day-night-paired tows with a MOCNESS (Multiple Opening/Closing Net and Environmental Sensing System)[67]. Individual nets collected samples with ~50-m depth resolution from the surface to a depth of 450 m[68]. Samples were preserved in 1.8% buffered formalin and then scanned using a ZooScan imaging device, sorted into broad taxonomic groups using Deep Learning methods[69] followed by 100% manual validation, then analyzed for morphometric information (e.g., length, width)[70]. Abundance was converted to biomass using length-carbon relationships[30]. Four taxa had significantly elevated surface biomass during the night: copepods, euphausiids, chaetognaths, and 'others' (a group that was mostly hyperiid amphipods). To calculate active transport, we first calculated (for each net and taxon) the summed biomass within 11 logarithmically spaced size bins (from <0.2 mm to >10 mm). Then for each depth horizon (e.g., 100 m, 150 m, ... 450 m) and each taxon and size class, we quantified the night-day biomass difference vertically integrated above the depth horizon. We assumed that zooplankton that migrated deeper than 450 m resided at a depth between 450 and 600 m because few zooplankton migrate deeper than this depth[35]. Because euphausiids are known net avoiders and net avoidance is most likely during the day, we adjusted daytime vertically integrated euphausiid biomass to match nighttime vertically integrated biomass. We estimated the specific respiration rate of each taxon and size class using allometric scaling relationships given in ref. 71 for copepods and ref. 72 for other mesozooplankton. We assumed that dissolved organic carbon excretion was equal to 31% of respiration[8]. For further details of zooplankton active transport calculations, see ref. 30.

We used estimates of active transport mediated by DVM mesopelagic fish (mostly Myctophidae) during our Lagrangian experiments from refs. 32, 37. Briefly, day-night differences in fish biomass were quantified from results of day-night-paired Matsuda-Oozeki-Hu net trawls (5 $m^2$ opening, 1.7 mm mesh), and respiration of vertical migrants at depth was estimated using a bioenergetics model (see ref. 32 for additional details). DVM fish daytime residence depths were assumed to be normally distributed with a mean of 450 m and a standard deviation of 50 m based on ref. 33.

Complete active transport estimates were only available for two cruises (P0704 and P0810, with a combined nine Lagrangian experiments). However, these cruises both took place during El Nino-Southern Oscillation neutral years and covered two seasons (spring and autumn) while sampling spatial variability from coastal to oligotrophic regions. We thus consider them to be broadly representative of typical conditions across the CCE (for instance, the mean sediment-trap-derived sinking carbon flux on these nine Lagrangian experiments was 8.8 mmol C $m^{-2}$ $d^{-1}$, compared to a regional annual average ($\bar{F}_{100}$) of 9.0 mmol C $m^{-2}$ $d^{-1}$). Consequently, we use the mean of the active transport across these nine Lagrangian experiments as our regional average for active transport. We quantified uncertainty in this regional average using a nonparametric bootstrapping (sampling with replacement) approach. We note, however, that these estimates of active transport are unequivocally underestimates of the total magnitude of active transport because they only account for export mediated by the respiration and or excretion of taxa undergoing normal diel vertical migration. They thus do not account for taxa that undergo reverse DVM[73], ontogenetic vertical migration[9], or that undertake foraging forays[36]. They also do not account for defecation or mortality at depth. The latter, in particular, is very difficult to constrain but has been estimated (using a food-web model informed with mesopelagic taxa respiratory requirements, among other constraints) to be substantial in the CCE[37].

### Subduction

It is not possible to directly measure the physical transport of particulate organic carbon by the combined actions of the eddy subduction pump, mixed layer pump, and large-scale circulation. Hence we used a data-assimilation approach that combined a Lagrangian particle creation, remineralization, and sinking model (particle model[39]) with an ocean circulation model (9-km resolution, ROMS 4DVAR[74,75]). The particle model assimilated in situ sediment trap data, NPP, and vertical profiles of particulate organic carbon to objectively constrain particle production rates as a function of depth, particle sinking speeds for slowly (i.e., phytoplankton) and rapidly (i.e., fecal pellets) sinking particles (simulated using continuous particle sinking speed spectra), and particle remineralization rates. The particle model was then coupled to an ocean-circulation state estimate from the ROMS 4DVAR model to enable particle transport in three dimensions as a result of physical circulation and particle sinking. The ROMS 4DVAR state estimate assimilated in situ data from our cruise (temperature and salinity profiles, along with sea surface height and temperature from remote sensing) into dynamically consistent 2-month estimates of four-dimensional currents centered around our cruise dates. Within the combined models, we classified particles as "sinking" past a specific depth horizon if they passed that horizon during the particle model time step and "subducted" if they passed that horizon during the physical circulation model time step. We note that "subduction" here includes all physical transport processes (e.g., a combination of the mixed layer pump, eddy subduction pump, large-scale advection, and vertical mixing as defined in refs. 2,14). For further details on the joint modeling system used to quantify subduction, see ref. 39. Subduction estimates were only available for three cruises (P0605, P0704, and P0810). These cruises included a total of 13 Lagrangian experiments from two seasons (spring and autumn), and all occurred during El Niño-neutral conditions while sampling a range of conditions from coastal upwelling to oligotrophic offshore regions. We estimated regional average subduction rates as a function of depth from the arithmetic mean of these 13 Lagrangian experiments. We quantified uncertainty in the regional average using a nonparametric bootstrapping approach. We note that these estimates of subduction account only for the physical transport of particles (not dissolved organic matter) because our approach did not allow us to rigorously constrain dissolved organic matter remineralization rates.

### Sequestration time

To quantify the duration of time that remineralized $CO_2$ will be sequestered, we utilized an offline version of the MITgcm Lagrangian 'floats' package[76]. The model was forced with four-dimensional flow

fields from the data-assimilating global 1/12° HYCOM reanalysis (GOFS 3.1 GLBb0.08 expt_53X)[77], which spans 1994–2015. The global model domain was subsetted to 100 °E–76.08 W° and 0 °N–66.51 °N (Supplementary Fig. 1a). HYCOM flow fields were converted from hybrid coordinates to z-coordinates and daily-average velocities for use in the MITgcm model were computed using procedures developed in ref. 78. Forty-five vertical layers were used (10 m thickness from 0 to 150 m depth, 25 m thickness from 150 to 600 m depth, 50 m thickness from 600 to 1000 m depth, with the remaining layers at 2000 m, 4000 m, and the maximum depth of 5952 m). We released simulated passive floats (mimicking remineralized carbon export) at eight depths (100, 150, 200, 300, 400, 500, 750, and 1000 m) and five locations along a cross-shore transect in the CCE. Floats were released daily for the first 5 years of the simulation and tracked for 500 years (the 22 years of HYCOM velocity fields were looped cyclically, and hence results should be considered indicative of sequestration time based on circulation patterns from 1994 to 2015 rather than a forecast of sequestration times for particles sequestered today that are subsequently exposed to future, likely modified ocean circulation). Floats were tracked until they either (1) were entrained into the mixed layer, (2) reached a model boundary (typically the equator), or (3) until the end of the 500-year simulation. Mean sequestration time was quantified as:

$$g(z) = \frac{1}{n_z} \left( \sum_{i=1}^{n_{mld,z}} t_{mld,i} + \sum_{i=1}^{n_{f,z}} t_f + g(z_{t=t_{final}}) + \sum_{i=1}^{n_{b,z}} t_{b,i} + g(z_{t=t_{boundary,i}}) \right) \quad (4)$$

The three summation terms refer to the sequestration time for individual floats that reached the mixed layer during the 500-year simulation, the expected sequestration time of floats that remained in the domain but did not reach the mixed layer during the simulation, and the expected sequestration time for floats that hit a boundary during the simulation, respectively. $n_z$ is the number of floats released from depth $z$. $n_{mld,z}$, $n_{f,z}$, and $n_{b,z}$ represent the number of floats from a given depth that reached the mixed layer, remained sequestered, or hit a boundary, respectively ($n_z = n_{mld,z} + n_{f,z} + n_{b,z}$). $t_{mld,i}$, $t_f$, and $t_{b,i}$ equal the time elapsed before an individual float was entrained into the mixed layer, the entire time of the simulation, or the time it took for a particle to hit a boundary, respectively. Because the mean sequestration time is a function of the expected length of time for floats that did not reach the mixed layer during the simulation, we solved this equation iteratively, accounting for the updated mean when including floats that remain sequestered or hit the boundary. To smoothly interpolate between depths, we used a piecewise cubic polynomial function for $g(z)$. We quantified uncertainty in $g(z)$ based on the differences in mean sequestration time between different years of particle launch. For each BCP pathway, the amount of carbon sequestered in the ocean as a result of export occurring in the CCE domain was quantified by vertically integrating the product of remineralization as a function of depth (for that pathway) and sequestration duration ($g(z)$) as a function of depth. For carbon dioxide remineralized deeper than 1000 m (the deepest depth at which we released simulated floats), we assumed a sequestration duration equal to our estimated mean sequestration duration at 1000 m (1335 years). Regional carbon sequestration was further determined by multiplying by the area of the CCE domain as defined in ref. 25 and Fig. 5b.

### Reporting summary
Further information on research design is available in the Nature Portfolio Reporting Summary linked to this article.

## Data availability
In situ data used in this study and CCE monthly state estimates used for the subduction model are available through the CCE LTER Datazoo repository (https://oceaninformatics.ucsd.edu/datazoo/catalogs/ccelter/datasets), the Zooscan Database (https://oceaninformatics.

ucsd.edu/zooplankton/zooscandb), and/or the Environmental Data Initiative (https://doi.org/10.6073/pasta/de679918c44266dcebbc5f85 a37dcd36, https://doi.org/10.6073/pasta/d19f13b361177f7d10135d484 98fa7c0, https://doi.org/10.6073/pasta/4c90b0a9fca143c5203ba020 27030555, https://doi.org/10.6073/pasta/3c607138a88218846cae4d6 f201942f6, https://doi.org/10.6073/pasta/0a0d884667f55ffe551e33dcf 7ebe535). HYCOM data used can be obtained from the Naval Research Laboratory (https://www.hycom.org/dataserver/gofs-3pt1/reanalysis). Data are also available in Supplementary Tables S1–S3, Supplementary Data 1–3, and in a source data file that can be used to recreate figures. Source data are provided with this paper.

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

## Acknowledgements

We would like to thank the captains and crews of the Research Vessels Melville, Revelle, Knorr, Thompson, Atlantis, and Sikuliaq. We are also indebted to our numerous colleagues in the CCE LTER program, especially Mike Landry, Kathy Barbeau, Ralf Goericke, Mati Kahru, Art Miller, Hajoon Song, and Shonna Dovel. This research was supported by National Science Foundation grants OCE-0417616, OCE-1026607, OCE-1637632, OCE-1614359, and OCE-2224726 to the CCE LTER Program.

## Author contributions

M.R.S. was responsible for at-sea sediment trap measurements, along with T.B.K., N.Y., and C.K.F. M.D.O. contributed vertically stratified zooplankton data. J.P.I. conducted sequestration temporal horizon simulations. T.B.K. was responsible for remote-sensing syntheses of sinking particle flux. M.R.S. performed subduction simulations, conducted synthetic analyses, and developed regional budgets. All authors contributed to writing and editing the manuscript.

## Competing interests

The authors declare no competing interests.
