## [Peer Review File · Nature Communications]

Carbon sequestration by multiple biological pump pathways in a coastal upwelling biomeREVIEWER COMMENTS

Reviewer #1 (Remarks to the Author):

see attached files (pdf contains marked up specific comments)

Review of Stukel

General

The authors have an unprecedented opportunity to look at a range of particle pumps based on an extensive and high-quality data set from CCE. The data quality and assumptions used are well justified for BGP and MMP, but in contrast, are opaque for the PIP (subduction and vertical mixing). More details are needed on the particle types, their biogeochemical characteristics, and assumptions about their particle transformations as they were physically injected to depth.

The title and abstract are slightly misleading as the CCE is a heterogeneous region as the authors point out in the Introduction.

More information is needed on the seasonality of sampling across the many voyages to the CCE. This should include any sampling biases (i.e., focusing on one pump) and also the sampling across the three distinct biomes referred to in the Introduction. Information on the seasonality of the various pumps is provided in ref 4 (Boyd et al., 2019).

Also, there is little discussion or acknowledgment of how particles can 'jump pumps' for example will subducted particles be intercepted and swept to greater depths by gravitationally sinking particles? This lack of discussion can also relate to the double accounting of particles across pumps. Again see ref 4 for more details.

In the discussion, the link to climate change is a bit of a 'laundry list'. Surely they can use the seasonality of sampling and also their three biomes within the CCE to tease this debate (across different oceanic provinces and their main particle pumps) out further.

Apart from these mainly minor comments, this is another excellent study from this well-respected group. Publish after minor revisions.

Philip Boyd

Reviewer #2 (Remarks to the Author):

- What are the noteworthy results?

This study is the first of its kind to bring together observational data and modelling to determine the carbon export and sequestration of the 3 major biological pump pathways in a specific ocean region, i.e. the California Current Ecosystem. I commend the authors on bringing together a huge amount of observational data collected on many cruises and the integration and elevation of the data with various modelling approaches. I also commend the authors on the care and detail taken in quantifying the uncertainty of various calculations. I would recommend that the paper undergoes moderate revisions.

The authors find that the gravitational sinking flux dominates both export and sequestration in the CCE, whilst the physical pump makes the next most significant contribution to export and active transport makes the next most significant contribution to sequestration. This approach has allowed a complete view of the biological pump within the CCE whilst integrating over space, time and depth. The authors contextualise the results of the study in terms of how the biological pump might change due to climate feedbacks.

- Will the work be of significance to the field and related fields? How does it compare to the established literature? If the work is not original, please provide relevant references.

I believe this piece of work could be a significant contribution to the field after some suggested revisions. There is very little literature to compare it to (highlighting the novelty and huge effort put in by the authors) but it generally falls in with the review paper findings by Boyd et al. (2019) which shows that the gravitational pump dominates export and sequestration, the mesopelagic migrant pump (active transport) is the next largest contributor and the physical pumps make smaller contributions, especially to long term sequestration.

- Does the work support the conclusions and claims, or is additional evidence needed?

The work does support the conclusions but I have some queries and suggested revisions which I have outlined below. I would also encourage the authors to provide more detail in the methodology section and add some more detail surrounding important caveats to the approach in the main body of the text (i.e. active transport does not include fecal pellet production at depth or migrant mortality). I would like to see more information provided about the data presented in Figure 1, particularly around sampling time/ location. I have outlined this further in my comments below.

- Are there any flaws in the data analysis, interpretation and conclusions? - Do these prohibit publication or require revision?

I support the concept and general approach and conclusions of the paper but there are some areas which require clarification and some moderate revisions. I have outlined these fully below but I encourage the authors to focus on variability in the observational data (temporal/spatial) and how it may impact the contribution of the three pathways, the sequestration time and budget calculations, expanding on some areas of the methodology and making sure the links to the data repositories are correct.

- Is the methodology sound? Does the work meet the expected standards in your field?

The work does meet the expected standards for the field and I can see that particular care has been taken in most areas i.e. calculating Martin's b and estimating uncertainty for different pathways. Some areas require further detail to make it easier for the reader to understand how certain calculations were carried out (more detail below).

- Is there enough detail provided in the methods for the work to be reproduced?

I feel that as it stands there is not enough detail to replicate some areas of the study – in particular how the sinking vs subduction pathways were attributed in the model and how the sequestration time was calculated. In the future, I would encourage the authors to make it more accessible for others to reproduce their work by providing analysis and model code and/or processed output.

Specific comments by line number:

Line 13: Referring to the BCP as a climate change feedback is ambiguous. The BCP is a natural process that can be perturbed by climate changes and may lead to climate feedbacks. I think more careful phrasing is needed here.

Line 20: As the units presented are C units and previously only carbon has been referred to I would suggest deleting 'dioxide'.

Line 48: I don't agree with this sentence. The study characterizes 3 types of environments that are found across the globe, and for the subtropical gyre environment it spans a large proportion of the global ocean, but there are still many different environments that make up the global ocean. I would reframe this sentence.

Line 51: When referring to CO₂ sequestration I find it useful for the reader to define a timescale. Sequestered for 100 yrs? for 1000 yrs?

Line 54: Whilst the sediment trap flux profiles look sensible have you, or have previous studies, carried out an assessment on the possible hydrodynamics biases in shallow waters? It's not possible to tell from figure 1 which profiles are from the coastal site and which are offshore (perhaps use different markers?) but they may have different upper ocean turbulence regimes. What is the shallowest water depth that the cruises traversed over? I find myself wondering how this impacts the shape and spread of the profiles and the mean sinking flux estimates. I think more information needs to be included in Figure 1 to allow the reader to evaluate possible differences based on location/ time of year. I appreciate you are trying to create an overall budget for the CCE but it is impossible for the reader (or a reviewer!) to assess what might drive the spread in fluxes, which is still of interest and could influence the mean values. My intuition would be that the contribution of the pathways would change depending on intra- and inter-annual and the onshore upwelling vs offshore location.

Line 58: 'By combining satellite-observed NPP with an empirical fit between sinking particle flux and contemporaneous NPP measured in situ²⁷, we determined that average regional carbon export via sinking particles was 9.0 ± 2.2 mmol C m⁻² d⁻¹' – Was the empirical fit a strong relationship? Can you provide any statistics? I would be nice to see more information about this in the supplementary material.

Line 62-63: Ref 5 also uses satellite data but the sentence implies it doesn't. I would reframe the sentence to group all the different approaches together to provide the range.

Line 64 (Figure 1): Whilst I appreciate that not all experiments would have data available for each pathway. Did you have data about each pathway from each of the three areas in the CCE? Again, I can't glean any information about this from the figure and so I wonder if each pathway is well-represented for each region. Also, from the methods, it seems that a good amount of effort has been put into estimating uncertainty for various calculations but there is no measurement or analytical uncertainty presented in Figure 1. If available, please add. Finally, for plot c) does the red line finish at ~43? If not please adjust the axis.

Line 64-66: This sentence highlights one of my main concerns. Is it reasonable to combine observations from a coastal upwelling area and offshore cruises to provide an average value when the paper states that the different areas span beyond the observed average range for the global ocean? From reading the methodology below it's clear that there are different amounts of observations for each pathway. How might this impact the mean and contribution of each pathway? Is it biased to one region that is better sampled than another? If a BCP budget was calculated for each of the 3 areas would it be very different? I am not necessarily asking the authors to do this but the possible variability that could arise needs some careful thought. I am open to agreeing that the approach in the paper is the correct one but I would like to see more justification.

Line 72: provide ranges for the temperature and oxygen concentration inline.

Line 91: add value to text for 'active transport was typically a factor of 2 lower'

Line 115: add a reference to the sentence. Perhaps Halfter et al. 2021?

Line 124-126: From Stukel et al. (2018) - 'Both parameterizations suggested that subduction is an important, at times dominant, mechanism of POC vertical export in the region (median 44% and 23% contribution to total POC export for PFP and Aggregate parameterizations at the 100-m depth horizon). The percentage contribution of subduction was highly variable across water parcels (ranging from 7% to 90%), with subduction typically more important in offshore, oligotrophic regions. On

average the fate of particles that are passively transported out of the surface layer by advection is different from that of particles that sink across the 100-m depth horizon. Subducted particles were predominantly remineralized shallower than 150 m, while approximately 50% of gravitationally exported POC was remineralized at depths >500 m.' Some of the cruise data used in Stukel et al. (2018) was also used here (as far as I can tell) and this excerpt from that paper highlights the variability across the different environments that have been averaged in this paper to give one value. I appreciate that the findings in relation to sequestration are consistent with Stukel et al. (2018) but as above I require more justification for creating one BCP budget.

Figure 3: Depending on the experiment active transport can account for 5-45%, sinking can account for 35-95% and mixing can account for 0-60%. This puts some interesting and useful bounds on these pathways. Is there even more information to be extracted besides depth? Could you use different markers depending on the region sampled? i.e. oligotrophic vs coastal upwelling? I also find myself wondering how this plot would look for the sequestered C at 1000m.

Line 133: This is an important point - perhaps earlier on it needs to be qualified that this focuses on particulate carbon sequestration. Also, what about suspended particles? Not all particles sink.

Line 139: 'most CO₂ molecules respired at 100m depth were entrained into the mixed layer' - can you quantify this? What %?

Line 140: the paper calculates mean sequestration time - for Lagrangian studies, Siegel et al. (2021) highlighted that the median is often a better metric. For example, a mean sequestration time of 102 +/- 3 years for particles at 100m when 'most' were rapidly entrained into the mixed layer seems odd. If you have >50% entrained rapidly (what is rapidly here? Within 1 year? 10 years? 50 years?) back into the mixed layer and 'some' were sequestered for centuries then how do you end up with a standard deviation of +/- 3 years? There are a lot of vague descriptors here that can mean different things than intended (i.e. rapidly, most, substantial). I encourage the authors to put numbers in place of these words to allow the reader to understand more clearly what they have found. I am happy to be shown where I am wrong here but the text requires refinement so as not to confuse the reader. I also find the standard deviations on all of the sequestration times quite low, especially when considering the generally chaotic nature of Lagrangian simulations.

Line 145: you refer to using the remineralization rate to determine the C sequestered by each pathway. This is the first reference to the remineralization rate. How was this quantified? Is it from observations or a modelled rate? Where this comes from and some indication of what the rates are needed to be available for the reader to evaluate.

Line 148: One important point that needs clarity is how the sequestration times can be longer than the 500-year simulation. This implies some assumption is made either at the edge of the model boundary or at the end of the 500 years. More detailed methodology is needed here, especially around what assumptions are made when a particle reaches the model boundary. What if a particle is just below the mixed layer when it reaches the model boundary? If it was allowed to travel further would it then enter the mixed layer? I encourage the authors to elaborate on this in the supplementary material.

Line 149-150 & Figure 4 & Table 1: I am having trouble reconciling the sequestration times and the regional sequestration estimates. The supplementary material says 'For each BCP pathway, the amount of carbon sequestered in the ocean as a result of export occurring in the CCE domain was quantified by vertically integrating the product of remineralization as a function of depth (for that pathway) and sequestration duration ($g(z)$) as a function of depth.' I understand this in principle but a reader needs clarification about what values are depth-integrated. Are the orange values in Figure 4 between 100m and 400m, and between 400m and 1000m integrated over depth, whilst the orange value below 1000m isn't? Assuming that the 1.7PgC is not integrated over depth? I have tried to calculate how the 1.7 mmol C m⁻² d⁻¹ translates to the storage of 1.7 Pg C but I can't figure it out. I'm not sure how the authors handle the time component here and would appreciate it if the authors could elaborate. I am happy to be shown how I'm wrong here but it really is not clear for the reader to determine how these numbers have been reached.

Figure 4: a note needs to be made on the figure/ in the caption that active transport is 0 at 1000m because fecal pellet production deeper than 450m is not considered and neither is mortality. I know this is highlighted in the text but for those readers who skim the figures, this is an important caveat.

Line 156-196: These three paragraphs are currently organised as three distinct paragraphs with a literature review, contextualising findings from the study and how the BCP could change elsewhere in relation to each pathway. This makes it hard to follow as a reader with the text constantly jumping between the three pathways. I would suggest integrating the information for each pathway into a paragraph to improve the flow for the reader.

Line 316: reference 46 has some repetition and doesn't appear to have a title.

Line 335: Are the effects of different modes of the El Nino Southern Oscillation considerable at the CCE site? Was the sampling equally weighted for different El Nino modes?

Line 399-410: I appreciate that this is a huge amount of work and that it is necessary to make lots of assumptions i.e. abundance to biomass, specific respiration rate, DOC excretion. I think it is essential to highlight these assumptions briefly in the main body of the text and how this might impact the active transport fluxes/ sequestration budgets. The addition of the active transport pathway is the dominant advance from previous studies (e.g. Stukel et al. 2018) and so I think it requires more discussion on the caveats for those who might not delve into the methodology section.

Line 428: What are foraging sorties? The reference doesn't include the word sorties as far as I could tell. Is this a typo?

Line 434: Add model resolution.

Line 437: Where do the remineralisation rates come from? Apologies if I've missed it but I don't recall this being mentioned.

Line 442-444: What are the particle model and the physical circulation model timesteps? Are the models run so that, for example, the circulation model progresses (let's say with a 1 day timestep) and then the particle model is run within the 1 day (let's say with a 1 hour timestep)? If a particle is moved across the boundary within the 1st hour of the 1 day timestep how do you attribute it to either sinking or subduction? More detail in the text is required here.

Line 448: Am I correct that all active transport and subduction estimates were from El Nino-neutral conditions but that it is not true for the sinking flux estimates? Some comment on how this may impact the averages and contributions from the 3 pathways could be included.

Line 463: I'm not sure about the phrasing of 'remineralized carbon dioxide' – remineralized carbon export or dissolved inorganic carbon would be more precise. Also, see studies that have taken a similar approach and perhaps should be referenced Robinson et al. (2014) and Baker et al. (2022).
<https://agupubs.onlinelibrary.wiley.com/doi/full/10.1002/2013GL058799>
<https://agupubs.onlinelibrary.wiley.com/doi/pdf/10.1029/2021GB007286>

Line 490: It is great that the observational data is available (first and last hyperlink in the file also includes the), which breaks the link and the links are the same). It isn't clear what data is available or how to find it in the edirepository. I searched CCE LTER with no results. Linking to the exact datasets would be much more useful for the reader. The naval research laboratory link is the same as the zooscan database which I assume is a typo. In the future, the authors might consider making the model output available (e.g. zenodo is a free archive) or providing scripts to allow others to replicate the various complex modelling stages that have come together to produce the BCP pathway budgets. As it stands I think it would be rather difficult to replicate this aspect of the work.

Reviewer #3 (Remarks to the Author):

This manuscript details a large dataset collected over multiple field seasons that characterises the flux of organic carbon by 3 BCP pathways – gravitational sinking, active flux and physical pumps. The analysis seems fairly robust and provides useful conclusions regarding the relative contribution of the different pathways to total flux. However, the authors need to be clearer that these results are only valid for this particular upwelling-dominated region and can't necessarily be viewed as representative of the situation in other parts of the global ocean.

General comment:

The cruises cover several years and times of year, but this temporal variability is ignored for this analysis. That's fine because it's the best that can be done with the data available at the moment, but

I urge the authors to add a paragraph to the discussion on the likely temporal (seasonal) variability in the relative strength of the pathways and the implications of ignoring temporal variability for their results.

There are very large uncertainties in some of the calculations, particularly for the active fluxes. The implications of these large uncertainties and the validity of the underlying assumptions are not currently discussed in the manuscript (and should be). There are also aspects of the active flux not included in the calculations (as described on lines 424-430) which I think should be mentioned in the main body of the manuscript.

Complete active flux estimates are only available for 2 (of 11) cruises, and subduction estimates are only available for 3 (of 11) cruises. The authors need to include some discussion of whether these limited measurements are representative of the conditions in the region, and the influence of interannual/seasonal variability on the interpretation of their results as a consequence.

Having written out these general comments, it seems to me that a separate section is required in the manuscript which covers the assumptions, limitations and uncertainties of the calculations made.

Specific comments:

Line 14: not 'all BCP pathways' are quantified simultaneously here either (lipid pump not included, for example)

Line 20: do the authors truly mean 'carbon dioxide' here, or would 'organic C' be more precise?

Line 48-49: I think it's stretching credibility a little to claim that the region 'spans much of the ecosystem variability found in the global ocean'

Line 51: please include a map of the study region and the sampling regime. It would also be helpful to have an indication (supplementary table?) of the dates and year of each of the cruises.

Line 51 & onwards: reference to the Supplementary Methods needs to be made throughout

Line 60: what is the export depth in this study? Is it uniformly 100m?

Line 77-108: is the active flux remineralisation in Fig 2 the sum of DVM organisms' respiration and excretion?

Line 138-143: in the calculations of sequestration potential, do the tracked particles have a decay rate (respiration rate) as they travel? Or is the assumption that the POC remains within the water mass it entered on export until that water mass returns to the surface?

Line 147: is the higher sequestration rate for sinking particles due also to a slower remineralisation rate?

Line 443-444: what are the time steps for the particle model and physical model?

Reviewer #1

General The authors have an unprecedented opportunity to look at a range of particle pumps based on an extensive and high-quality data set from CCE. The data quality and assumptions used are well justified for BGP and MMP, but in contrast, are opaque for the PIP (subduction and vertical mixing). More details are needed on the particle types, their biogeochemical characteristics, and assumptions about their particle transformations as they were physically injected to depth.

We have added a supplementary section that gives substantial additional details about the approach used for quantifying subduction and vertical mixing.

The title and abstract are slightly misleading as the CCE is a heterogeneous region as the authors point out in the Introduction.

It is certainly not our intent to imply that our results come only from the portion of our study area that experiences active upwelling. However, the terms “coastal upwelling ecosystem” and “coastal upwelling biome” are often used to refer to the entire ecosystem that is influenced by upwelling (a point that we attempt to make much clearer in our added section “Spatiotemporal variability in BCP pathways in the CCE”. We have not added additional text to the title or abstract, because we are limited by word limits.

More information is needed on the seasonality of sampling across the many voyages to the CCE. This should include any sampling biases (i.e., focusing on one pump) and also the sampling across the three distinct biomes referred to in the Introduction. Information on the seasonality of the various pumps is provided in ref 4 (Boyd et al., 2019).

We have added new panels to Fig. 1 that show the seasonal coverage of our sampling program. In the CCE temporal variability in the different BCP pathways are likely driven mostly by changes in upwelling intensity. While mean upwelling intensity certainly shows seasonality (peaking in spring), in this region it can occur at any time of year and there is substantial variability across other time scales as well (e.g., interannual variability driven by ENSO and short term variability driven by episodic phenomena). Because we do not have enough sampling of the different pathways to define seasonal averages, we instead focus our discussion on the relationship between export and ecosystem properties that likely covary with upwelling intensity.

Also, there is little discussion or acknowledgment of how particles can ‘jump pumps’ for example will subducted particles be intercepted and swept to greater depths by gravitationally sinking particles? This lack of discussion can also relate to the double accounting of particles across pumps. Again see ref 4 for more details.

We have several sentences near the end of the section “An interconnected BCP in a changing ocean” that are focused on the ways in which carbon transported to depth via one BCP pathway can be transported further into the ocean’s interior via other pathways.

“We must also consider the inter-connectedness of the BCP pathways. While our conceptual model differentiates three processes it necessarily simplifies the complexity of the BCP. For instance, vertical migrators with long gut turnover times will defecate a portion of their ingested prey at depth (active transport) where it will be released as fecal pellets that contribute to further downward sinking flux.

Conversely, sinking particles can be consumed by vertical migrators that transport the carbon deeper into the ocean. Organic molecules cleaved from sinking particles by microbial hydrolytic exoenzymes⁶¹ may be subducted further into the ocean interior while subducted particles can aggregate in the deep ocean to form sinking marine snow⁶². Slowly-sinking particles in regions of active subduction or mixed layer deepening contribute simultaneously to multiple BCP pathways⁴³.”

While there are certainly more processes that could be considered, we do not believe it necessary to include more text here. It is very difficult to quantify the processes that lead to particles “jumping pumps” and our focus on this manuscript has been the aspects of the BCP that we were able to quantify.

In the discussion, the link to climate change is a bit of a ‘laundry list’. Surely they can use the seasonality of sampling and also their three biomes within the CCE to tease this debate (across different oceanic provinces and their main particle pumps) out further.

We have added a new section “Spatiotemporal variability in BCP pathways in the CCE” that uses spatiotemporal variability in the CCE to attempt to address these questions.

Apart from these mainly minor comments, this is another excellent study from this well-respected group. Publish after minor revisions. Philip Boyd

We thank the reviewer for his useful comments.

Reviewer #1 (additional comments embedded in the pdf):

indicate the seasonal coverage which is potentially important for different pumps

We have added panels to Fig. 1 that show the seasonal coverage.

sub-mesoscale eddy subduction?

The resolution of the data-assimilating ROMS 4DVAR state estimates are 9-km. This is not sufficient to detect submesoscale dynamics.

but not from all PIPs

The reviewer is referring to “particle injection pumps”, which is terminology from Boyd et al. (2019). While we do not define PIPs in exactly the same way as Boyd et al. (e.g., our “subduction + vertical mixing” is the sum of what they term “eddy-subduction”, “large-scale physical”, and “mixed-layer”, because it is difficult to precisely differentiate these processes in the ocean), we do quantify all the PIPs that are *important* in the CCE. Notably, we do not quantify what they refer to as the “lipid pump”, but which we refer to as “active transport due to ontogenetic vertical migration”, because ontogenetic vertical migrators are not dominant taxa in the CCE (they are much more common in polar and subpolar regions).

Justify with a citation please.

The reviewer requests that we justify the statement that the CCE “spans much of the ecosystem variability found in the global ocean”. Please note that we have slightly modified this statement

by changing “ecosystem variability” to “productivity variability” to more accurately reflect the fact that ecosystems vary along many axes, while we are specifically referring to the fact that variability in productivity within the CCE is almost as variable as within the world ocean. While we do not have a specific citation for this fact, we present the following figure (for review only). It shows a heat map of sea surface temperature vs sea surface chlorophyll for the global ocean from the 1/12 degree SEAWIFS monthly climatology. Superimposed on this map, we include the specific sea surface temperatures and sea surface chlorophylls of our Lagrangian experiments (red diamonds), showing that our study locations encompass much of the global variability in sea surface chlorophyll.

interesting that despite the wide range of biomes referred to on lines 47 we see little variance - is this because one biome had a much larger areal extent than others?

The value given is an estimate of annual mean carbon export in our study domain plus or minus our uncertainty associated with this value. Thus it accounts for within-region variability, but is not an estimate of the variability between different regions of our study domain.

can you add some other notation - such as different dashed lines to indicate which of the biomes the voyages took place in - and also what seasons

We have added three new panels to give readers better context about the spatial and temporal variability in our sampling.

also net avoidance by macrozoo or fish etc

We have added the caveat “and because biomass can be underestimated when actively swimming taxa avoid nets”.

what about moderately sinking particles if its a spectrum

Yes, moderately sinking particles are certainly included as well, we just found it clunky to say “rapidly, moderately, and slowly sinking particles”. We hope that we have made this all more clear by adding our supplementary methods, which give far more detail on our approach for estimating carbon subduction rates.

Which data?

This is now explained in our supplementary methods section, but it includes vertical profiles of NPP and POC, sinking particle export measurements, and zooplankton gut pigment measurements.

What biogeochemical characteristics did you assign these particles? What about the interplay between the different pumps??

All particles had an associated settling velocity and remineralization rate. Our model explicitly accounts for interplay between the sinking particle pump and physical particle pumps (i.e., mixed layer pump + eddy subduction pump + large scale physical pump). We have given more details in our supplementary methods section.

how did you assess this??

These were quantified using our particle subduction model, which utilized a three-dimensional Lagrangian model (with a ROMS 4DVAR state estimate providing the circulation) in which each particle had a settling velocity and remineralization rate, both of which were constrained using field measurements. Details are now given in our supplementary methods section.

do you consider Co2 equilibration here?

No, this calculation refers to the amount of CO2 equivalents that are sequestered (i.e., the amount of carbon dioxide that has been removed from the atmosphere as a result of these processes in the CCE).

what particle categories dominated in the traps? what particle types did you subduct? vertically mix?

The sediment traps are dominated by fecal pellets in high biomass times/locations. In oligotrophic conditions they are dominated by unrecognizable material (e.g., old material that has likely been re-worked as it sank). We did not specifically subduct or vertically mix a class of

particles, but rather created 10,000 particles per cycle defined by settling velocities drawn from a continuous distribution. The continuous distribution was defined to mimic two classes of particles (phytoplankton and fecal pellets) both of which have continuous settling velocity distributions, but with distinctly different means. We have greatly expanded our supplemental methods section to include these details.

ballast effects but also alkalinity discount as they sink

The reviewer is referring to the role of sinking calcium carbonate in the “carbonate counterpump”. Sinking carbonate actually leads to a net flux of carbon dioxide from the ocean to the atmosphere. We make this clear by modifying the sentence to read: “Ocean acidification is influencing multiple trophic levels and most directly affecting calcifying organisms that contribute disproportionately to sinking flux while contributing to the carbonate counterpump⁵⁴”.

or type of particles being subducted and their flux attenuation

We believe that this comment is in reference to the line that states: “subduction/vertical mixing ... are likely underestimated due to our inability to constrain ... vertical transport of dissolved organic molecules.” We do not believe that adding this clause would be appropriate, because (as we make clear in the new draft with supplementary methods) we do not make explicit assumptions about the types of particles being subducted or their flux attenuation. Rather we define remineralization rates and continuous particle sinking speed spectra from the field data. Thus, flux attenuation is an emergent result constrained by the field data. While the importance of subduction will certainly vary depending on particle dynamics, there is no reason to assume that our approach underestimates particle subduction rates. However, as we state in this sentence, our approach certainly underestimates total organic carbon subduction rates, because it does not account for dissolved organic carbon, which we could not constrain from the field data.

but this is only one of three biomes in the CCE

Please see our new section which explains how lateral advection connects the three distinct biomes of the CCE, especially with respect to sinking and subducted particles.

discuss seasonality here also

Please see our previous comment about seasonality.

see also: 10.1029/2021GB007286

Thank you for pointing us to this recent citation. We now include it.

or interact with other pumps - such as differential sinking rates and being subsumed into the gravitational pump from the subduction pump - see ref 4

The particles in our subduction model already explicitly “jump pumps” in this way. They can be subducted past a specific depth horizon but continue to sink beyond a deeper depth horizon. We

now make this clear in our detailed supplementary section focused on the subduction methodology.

Reviewer #2 (Remarks to the Author):

- What are the noteworthy results?

This study is the first of its kind to bring together observational data and modelling to determine the carbon export and sequestration of the 3 major biological pump pathways in a specific ocean region, i.e. the California Current Ecosystem. I commend the authors on bringing together a huge amount of observational data collected on many cruises and the integration and elevation of the data with various modelling approaches. I also commend the authors on the care and detail taken in quantifying the uncertainty of various calculations. I would recommend that the paper undergoes moderate revisions.

The authors find that the gravitational sinking flux dominates both export and sequestration in the CCE, whilst the physical pump makes the next most significant contribution to export and active transport makes the next most significant contribution to sequestration. This approach has allowed a complete view of the biological pump within the CCE whilst integrating over space, time and depth. The authors contextualise the results of the study in terms of how the biological pump might change due to climate feedbacks.

We thank the reviewer for this assessment of our study.

- Will the work be of significance to the field and related fields? How does it compare to the established literature? If the work is not original, please provide relevant references.

I believe this piece of work could be a significant contribution to the field after some suggested revisions. There is very little literature to compare it to (highlighting the novelty and huge effort put in by the authors) but it generally falls in with the review paper findings by Boyd et al. (2019) which shows that the gravitational pump dominates export and sequestration, the mesopelagic migrant pump (active transport) is the next largest contributor and the physical pumps make smaller contributions, especially to long term sequestration.

- Does the work support the conclusions and claims, or is additional evidence needed?

The work does support the conclusions but I have some queries and suggested revisions which I have outlined below. I would also encourage the authors to provide more detail in the methodology section and add some more detail surrounding important caveats to the approach in the main body of the text (i.e. active transport does not include fecal pellet production at depth or migrant mortality). I would like to see more information provided about the data presented in Figure 1, particularly around sampling time/ location. I have outlined this further in my comments below.

We have responded to the reviewer's specific comments below and thank the reviewer for their support.

- Are there any flaws in the data analysis, interpretation and conclusions? - Do these prohibit

publication or require revision?

I support the concept and general approach and conclusions of the paper but there are some areas which require clarification and some moderate revisions. I have outlined these fully below but I encourage the authors to focus on variability in the observational data (temporal/spatial) and how it may impact the contribution of the three pathways, the sequestration time and budget calculations, expanding on some areas of the methodology and making sure the links to the data repositories are correct.

We have responded to the reviewer's specific comments below.

- Is the methodology sound? Does the work meet the expected standards in your field?

The work does meet the expected standards for the field and I can see that particular care has been taken in most areas i.e. calculating Martin's b and estimating uncertainty for different pathways. Some areas require further detail to make it easier for the reader to understand how certain calculations were carried out (more detail below).

We have responded to the reviewer's specific comments below.

- Is there enough detail provided in the methods for the work to be reproduced?

I feel that as it stands there is not enough detail to replicate some areas of the study – in particular how the sinking vs subduction pathways were attributed in the model and how the sequestration time was calculated. In the future, I would encourage the authors to make it more accessible for others to reproduce their work by providing analysis and model code and/or processed output.

We have now included processed output as a supplement to the manuscript. We are not, however, able to archive the raw particle trajectories because of the large file sizes involved (e.g., the particle trajectories file for the subduction model alone is >100GB). While we generally agree with making code freely available (see, e.g., lead author Stukel's extensive publicly available model code on GitHub), we do not think it would be helpful in this case, because our model code used for diagnosing subduction (which is only a slightly modified version of the publicly available LTRANS code) is specifically designed to work with ROMS output files that are several gigabytes in size and hence too large for us to host with the manuscript.

Specific comments by line number:

Line 13: Referring to the BCP as a climate change feedback is ambiguous. The BCP is a natural process that can be perturbed by climate changes and may lead to climate feedbacks. I think more careful phrasing is needed here.

The reviewer is correct that the BCP is not inherently a climate change feedback, although any change that occurs in the BCP as a result of climate change would be expected to be a climate change feedback. We have rephrased to say: "However, our ability to predict future changes in the BCP – and whether these changes will serve as positive or negative climate feedbacks - is

hampered by the absence of studies that have simultaneously quantified all BCP pathways for any ocean region.”

Line 20: As the units presented are C units and previously only carbon has been referred to I would suggest deleting 'dioxide'.

Deleted.

Line 48: I don't agree with this sentence. The study characterizes 3 types of environments that are found across the globe, and for the subtropical gyre environment it spans a large proportion of the global ocean, but there are still many different environments that make up the global ocean. I would reframe this sentence.

The reviewer is correct, of course, that we do not actually span most of the *ecosystem* variability in the global ocean, because ecosystems vary in many different ways. However, the CCE does contain most of the variability in net primary productivity found in the global ocean, and hence we have modified the sentence to reflect this. (Modified sentence reads: “It <the CCE> thus spans much of the productivity variability found in the global ocean”. To support this assertion we have provided the following figure (for review only). It shows a heat map of sea surface temperature vs sea surface chlorophyll for the global ocean from the 1/12 degree SEAWIFS monthly climatology. Superimposed on this map, we include the specific sea surface temperatures and sea surface chlorophylls of our Lagrangian experiments (red diamonds), showing that our study locations encompass much of the global variability in sea surface chlorophyll. We also now include this sea surface temperature and sea surface chlorophyll data (along with other data) in Supp. Table 1.

Line 51: When referring to CO₂ sequestration I find it useful for the reader to define a timescale. Sequestered for 100 yrs? for 1000 yrs?

We are explicit about our time scales throughout the results section of the manuscript. However, because we make it clear that time scale varies for different BCP pathways and for different remineralization depths, we prefer not to specify a single time scale here.

Line 54: Whilst the sediment trap flux profiles look sensible have you, or have previous studies, carried out an assessment on the possible hydrodynamics biases in shallow waters? It's not possible to tell from figure 1 which profiles are from the coastal site and which are offshore (perhaps use different markers?) but they may have different upper ocean turbulence regimes. What is the shallowest water depth that the cruises traversed over? I find myself wondering how this impacts the shape and spread of the profiles and the mean sinking flux estimates. I think more information needs to be included in Figure 1 to allow the reader to evaluate possible differences based on location/ time of year. I appreciate you are trying to create an overall budget for the CCE but it is impossible for the reader (or a reviewer!) to assess what might drive the spread in fluxes, which is still of interest and could influence the mean values. My intuition would be that the contribution of

the pathways would change depending on intra- and inter-annual and the onshore upwelling vs offshore location.

First, with respect to potential hydrodynamic biases: We have addressed this issue directly in previous papers. We consistently utilize ^{238}U - ^{234}Th disequilibrium as an independent estimate of sinking particle flux. The agreement (and occasional disagreement) between sediment trap-measured flux and ^{238}U - ^{234}Th disequilibrium-derived flux is discussed extensively in Stukel et al. (2013, MEPS), Morrow et al. (2018, DSR), and Stukel et al. (2019, Mar. Chem.), so we prefer not to repeat the discussion here, but the summary is that there is no evidence for hydrodynamic biases. The two measurements usually agree to within the error bar and when there is a substantial disagreement, the broader evidence suggests that the cause of the disagreement is not related to sediment trap inaccuracy but rather a violation of the non-steady state without upwelling assumption that is typically applied for ^{238}U - ^{234}Th disequilibrium (in other words, the periods with substantial disagreement were typically during upwelling events that introduced low thorium water into the euphotic zone, thus leading the thorium approach to underestimate flux). The figure above on the right is from Stukel et al. (2019, Mar. Chem. 212: 1-15. doi: 10.1016/j.marchem.2019.01.003).

We agree with the reviewer that spatiotemporal variability in ecosystem processes likely shapes variability in the relative magnitudes of the three BCP pathways quantified in our study. We had addressed (albeit briefly) how differences in ecosystem structure between our cruises (and the fact that we did not sample all pathways on all cruises) might impact our results by noting that the mean sinking particle exports (at 100 m depth) on the experiments for which we quantified active transport and subduction were very close to the regional mean sinking particle export at 100 m depth. However, we can see that this is not sufficient information to really assess the impacts of spatiotemporal variability on our results. Hence we have substantially revised the manuscript to address this topic. Specifically, we made two major changes. First, we added three new panels to Figure 1 that show: spatial variability in net primary productivity along with the sampling locations for each cruise (a), temporal variability in NPP along with the times of each cruise (b), and temporal variability in the ENSO index along with the times of each cruise (c). Second, we added a new section entitled “Spatiotemporal variability in BCP pathways in the CCE” that

addresses these issues in detail. We have also added a supplemental table (Supp. Table 1) that gives contextual data (temperature, nitrate, chl, NPP) for all Lagrangian experiments.

Line 58: 'By combining satellite-observed NPP with an empirical fit between sinking particle flux and contemporaneous NPP measured in situ²⁷, we determined that average regional carbon export via sinking particles was $9.0 \pm 2.2 \text{ mmol C m}^{-2} \text{ d}^{-1}$ ' – Was the empirical fit a strong relationship? Can you provide any statistics? I would be nice to see more information about this in the supplementary material.

We now give additional details in the methods: “To determine a regional, annual estimate of sinking particle flux, we previously developed a relationship between *in situ* sinking carbon flux, NPP, and temperature²⁸: Sinking particle flux = $e\text{-ratio} \times \text{NPP}$, where $e\text{-ratio} = 0.056 (\pm 0.008) \times \text{SST} - 0.698 (\pm 0.122)$, $R^2 = 0.67$, $p < 0.001$.”

Line 62-63: Ref 5 also uses satellite data but the sentence implies it doesn't. I would reframe the sentence to group all the different approaches together to provide the range.

We understand how this phrasing could have been misleading and have re-phrased to: “For comparison, global satellite-derived estimates of sinking particle flux predict ocean-average values of $\sim 3 \text{ mmol C m}^{-2} \text{ d}^{-1}$ at the 100-m depth horizon⁸ and $3 - 8 \text{ mmol C m}^{-2} \text{ d}^{-1}$ at the base of the euphotic zone^{6,7,29,30}.”

Line 64 (Figure 1): Whilst I appreciate that not all experiments would have data available for each pathway. Did you have data about each pathway from each of the three areas in the CCE? Again, I can't glean any information about this from the figure and so I wonder if each pathway is well-represented for each region.

Yes, we have at least two Lagrangian experiments conducted in each region for each pathway.

Also, from the methods, it seems that a good amount of effort has been put into estimating uncertainty for various calculations but there is no measurement or analytical uncertainty presented in Figure 1. If available, please add.

Our goal in this figure is to show variability in each process with depth and across different Lagrangian experiments. While we understand the value of showing analytical uncertainty, in practice we have found that attempting to display analytical uncertainty on this figure makes the figure too messy such that viewers can no longer easily take away the salient points. Furthermore, the analytical uncertainty is generally much smaller than the variability between experiments.

Finally, for plot c) does the red line finish at ~ 43 ? If not please adjust the axis.

We have slightly modified the axes to make the surface value for 0605-1 clear.

Line 64-66: This sentence highlights one of my main concerns. Is it reasonable to combine observations from a coastal upwelling area and offshore cruises to provide an average value when the paper states that the different areas span beyond the observed average range for the global ocean? From reading the methodology below it's clear that there are different amounts of observations for each pathway. How might this impact the mean and contribution of each pathway? Is it biased to one region that is better sampled than another? If a BCP budget was calculated for each of the 3 areas would it be very different? I am not necessarily asking the authors to do this but the possible variability that could arise needs some careful thought. I am open to agreeing that the approach in the paper is the correct one but I would like to see more justification.

We understand the rationale behind this critique and we carefully thought through these issues when designing our synthesis approach. In short, we believe it is necessary to quantify the mean contribution for each pathway across the region, because it is an interconnected region linked through horizontal currents. In other words, it would not be appropriate to treat the ecosystem as a one-dimensional system in which the BCP operates in each region without influence from the others. We hope that we have made this much more clear in the new section "Spatiotemporal variability in BCP pathways in the CCE".

Line 72: provide ranges for the temperature and oxygen concentration inline.

Added.

Line 91: add value to text for 'active transport was typically a factor of 2 lower'

We are not entirely sure what the reviewer is referring to here. We give the range, mean, and confidence interval for fish-related active transport here.

Line 115: add a reference to the sentence. Perhaps Halfter et al. 2021?

Citation added.

Line 124-126: From Stukel et al. (2018) - 'Both parameterizations suggested that subduction is an important, at times dominant, mechanism of POC vertical export in the region (median 44% and 23% contribution to total POC export for PFP and Aggregate parameterizations at the 100-m depth horizon). The percentage contribution of subduction was highly variable across water parcels (ranging from 7% to 90%), with subduction typically more important in offshore, oligotrophic regions. On average the fate of particles that are passively transported out of the surface layer by advection is different from that of particles that sink across the 100-m depth horizon. Subducted particles were predominantly remineralized shallower than 150 m, while approximately 50% of gravitationally exported POC was remineralized at depths > 500 m.' Some of the cruise data used in Stukel et al. (2018) was also used here (as far as I can tell) and this excerpt from that paper

highlights the variability across the different environments that have been averaged in this paper to give one value. I appreciate that the findings in relation to sequestration are consistent with Stukel et al. (2018) but as above I require more justification for creating one BCP budget.

We have added the section: “Spatiotemporal variability in BCP pathways in the CCE” to address this question.

Figure 3: Depending on the experiment active transport can account for 5-45%, sinking can account for 35-95% and mixing can account for 0-60%. This puts some interesting and useful bounds on these pathways. Is there even more information to be extracted besides depth? Could you use different markers depending on the region sampled? i.e. oligotrophic vs coastal upwelling? I also find myself wondering how this plot would look for the sequestered C at 1000m.

As suggested, we have modified the figure to show different symbols for different regions.

With regards to the question about export flux across the 1000-m depth horizon, the figure is not particularly interesting because at this depth horizon export is almost completely dominated by sinking flux (as we indicate in Fig. 4).

Line 133: This is an important point - perhaps earlier on it needs to be qualified that this focuses on particulate carbon sequestration. Also, what about suspended particles? Not all particles sink.

We agree that this is an important point, and note that while we first specifically make this distinction at this line, it is also clearly indicated at the beginning of the subduction section when we state that “We estimate POC subduction and vertical mixing rates by combining...” We do not want to confuse any potential readers, however, and hence have also modified the abstract to clarify that these estimates apply to “subduction + vertical mixing *of particles*” (italicized portion is newly added). Please note that this calculation explicitly includes all particles (sinking and suspended).

Line 139: ‘most CO₂ molecules respired at 100m depth were entrained into the mixed layer’ – can you quantify this? What %?

The following is a histogram of the length of time for molecules to reach the mixed layer for different depths of release:

Line 140: the paper calculates mean sequestration time – for Lagrangian studies, Siegel et al. (2021) highlighted that the median is often a better metric. For example, a mean sequestration time of 102 +/- 3 years for particles at 100m when ‘most’ were rapidly entrained into the mixed layer seems odd. If you have >50% entrained rapidly (what is rapidly here? Within 1 year? 10 years? 50 years?) back into the mixed layer and ‘some’ were sequestered for centuries then how do you end up with a standard deviation of +/- 3 years? There are a lot of vague descriptors here that can mean different things than intended (i.e. rapidly, most, substantial). I encourage the authors to put numbers in place of these words to allow the reader to understand more clearly what they have found. I am happy to be shown where I am wrong here but the text requires refinement so as not to confuse the reader. I also find the standard deviations on all of the sequestration times quite low, especially when considering the generally chaotic nature of Lagrangian simulations.

The median is not necessarily a “better” metric than the mean. The median and the mean simply convey different information and are appropriate for different purposes, depending on the shape of the underlying frequency distribution. We could not find any part of the Siegel et al. paper where the authors state that the median is better than the mean, although it can be assumed that the authors prefer the median as they default to using this metric in some of their figures. However, what they specifically state is that metrics beyond the mean and median sequestration time should be considered (e.g., “the fraction of CO₂ injected that remains in the ocean over a planning time horizon”).

Towards this final point, it is important to note that the purpose of our study and the Siegel et al. study are subtly but distinctly different. The Siegel et al. study was focused on quantifying the efficacy of different locations as sites for carbon-injection projects. Thus the relevant question

was: If carbon is injected at a specific site and depth, how long will this carbon be removed from the atmosphere and more specifically how much of the carbon will remain sequestered over specific temporal horizons?

For our study, the focus was on the BCP's natural ability to sequester carbon dioxide and hence we were asking the question, how much carbon is currently stored in the ocean as a result of BCP activity in the California Current Ecosystem? Thus there is no specific temporal horizon that we are investigating nor is there a specific injection depth, but instead carbon dioxide that is remineralized at all depths.

When discussing the applicability of the median vs. the mean, it is important to carefully keep in mind the goal of the study. For Lagrangian studies, the median is often a more *robust* metric, because with highly skewed distributions rare particles with a long sequestration time can have a large influence on the mean. This is particularly problematic if either a small number of particles are used or if the duration of the simulation is short (this is why we released a large number of particles (73,000) and ran the simulation for a long time (500 years)). However, the median does not quantify the *expected* length of time that a carbon dioxide molecule will be sequestered. For that purpose the mean is by definition the more appropriate metric. This is highlighted with a simple hypothetical example: Consider a species of copepod species that vertically migrates to a depth of 200 m in an oceanic region that is generally downwelling, but has a maximum deep winter mixed layer of 220 m. When that copepod respire at 200 m depth in the summer, the carbon dioxide molecules will get transported vertically and horizontally by small scale motions, with likely half of the molecules transported upwards and half transported downwards. Because the initial remineralization depth of the molecules is shallower than the winter mixed layer, over half of these molecules will likely get respired within a year. Thus the median sequestration time will be less than a year (probably less than half a year). However, if we consider the molecules that were (through random water motions) transported to depths deeper than 220 m before the onset of the deep winter mixing, these molecules will likely remain in the ocean for longer than a year, and in fact, since this is a region of generally downwelling circulation, many of them will remain in the ocean for many years, decades, or centuries. If even 10% of these molecules are transported downward and sequestered for 500 years, the mean sequestration time will thus be >50 years, even though the median sequestration time is on the order of half of a year. Now consider the question (relevant to our study) of how much carbon dioxide is actually stored in the ocean as a result of the vertical migration behavior of this copepod. Let's assume that it respire $1 \text{ mmol C m}^{-2} \text{ d}^{-1}$. If we made the mistake of calculating the total carbon stored in the ocean due to this copepod using the median sequestration time, we would multiply $1 \text{ mmol C m}^{-2} \text{ d}^{-1}$ by 180 days and find that $180 \text{ mmol C m}^{-2}$ are stored in the ocean. This is equivalent to stating that the amount of carbon stored in the ocean as a result of this copepod's activity is equivalent to 180 days worth of its respiration. However, we know that this is not accurate. Since 10% of the carbon dioxide respired by this copepod will remain at depth for 500 years, we know that 10% of the respiration done by copepods 500 years ago remains sequestered in the ocean. Thus the expected length of time that a molecule of carbon dioxide is sequestered is greater than 50 years and the amount of carbon sequestered by this species has to be greater than $1 \text{ mmol C m}^{-2} \text{ d}^{-1} \times$

18250 days = 18250 mmol C m⁻². This is why the mean is the appropriate metric for our study, even though it requires both a longer model run and substantially more simulated particles released in order to get a robust estimate.

With regards to the uncertainty estimates, we are not computing the uncertainty in the length of time that an individual molecule is likely to be sequestered, but rather uncertainty in the expected length of time that all molecules respired at a specific depth are likely to remain sequestered. We calculate this from the standard deviation of the expected sequestration time as calculated for particles released during different years. This is stated in the methods section: “We quantified uncertainty in $g(z)$ based on the differences in mean sequestration time between different years of particle launch”

Line 145: you refer to using the remineralization rate to determine the C sequestered by each pathway. This is the first reference to the remineralization rate. How was this quantified? Is it from observations or a modelled rate? Where this comes from and some indication of what the rates are needed to be available for the reader to evaluate.

This comes directly from the data presented in Figs. 1c-d and 2. The remineralization rate (shown in Fig. 2) is the derivative of flux with depth. We now make this clear in the text.

Line 148: One important point that needs clarity is how the sequestration times can be longer than the 500-year simulation. This implies some assumption is made either at the edge of the model boundary or at the end of the 500 years. More detailed methodology is needed here, especially around what assumptions are made when a particle reaches the model boundary. What if a particle is just below the mixed layer when it reaches the model boundary? If it was allowed to travel further would it then enter the mixed layer? I encourage the authors to elaborate on this in the supplementary material.

This is explained in the current draft of the manuscript: . Mean sequestration time was quantified as:

$$g(z) = \frac{1}{n_z} \left(\sum_{i=1}^{n_{mld,z}} t_{mld,i} + \sum_{i=1}^{n_{f,z}} t_f + g(z_{t=t_{final}}) + \sum_{i=1}^{n_{b,z}} t_{b,i} + g(z_{t=t_{boundary,i}}) \right)$$

The three summation terms refer to the sequestration time for individual floats that reached the mixed layer during the 500-year simulation, the expected sequestration time of floats that remained in the domain but did not reach the mixed layer during the simulation, and the expected sequestration time for floats that hit a boundary during the simulation, respectively. n_z is the number of floats released from depth z . $n_{mld,z}$, $n_{f,z}$, and $n_{b,z}$ represent the number of floats from a given depth that reached the mixed layer, remained sequestered, or hit a boundary,

respectively ($n_z = n_{mld,z} + n_{f,z} + n_{b,z}$). $t_{mld,i}$, t_f , and $t_{b,i}$ equal the time elapsed before an individual float was entrained into the mixed layer, the entire time of the simulation, or the time it took for a particle to hit a boundary, respectively. Because the mean sequestration time is a function of the expected length of time for floats that did not reach the mixed layer during the simulation, we solved this equation iteratively accounting for the updated mean when including floats that remain sequestered or hit the boundary. To smoothly interpolate between depths, we used a piecewise cubic polynomial function for $g(z)$.

Line 149-150 & Figure 4 & Table 1: I am having trouble reconciling the sequestration times and the regional sequestration estimates. The supplementary material says ‘For each BCP pathway, the amount of carbon sequestered in the ocean as a result of export occurring in the CCE domain was quantified by vertically integrating the product of remineralization as a function of depth (for that pathway) and sequestration duration ($g(z)$) as a function of depth.’ I understand this in principle but a reader needs clarification about what values are depth-integrated. Are the orange values in Figure 4 between 100m and 400m, and between 400m and 1000m integrated over depth, whilst the orange value below 1000m isn’t? Assuming that the 1.7PgC is not integrated over depth? I have tried to calculate how the 1.7 mmol C m⁻² d⁻¹ translates to the storage of 1.7 Pg C but I can’t figure it out. I’m not sure how the authors handle the time component here and would appreciate it if the authors

could elaborate. I am happy to be shown how I’m wrong here but it really is not clear for the reader to determine how these numbers have been reached.

Our apologies. We believe that the reviewer is correctly interpreting what we did with the one key exception that we forgot to mention how we were treating remineralization of carbon that occurred deeper than 1000 m. For carbon dioxide remineralized deeper than 1000 m (the deepest depth at which we released simulated floats), we assumed a sequestration duration equal to our estimated mean sequestration duration at 1000 m (1370 years). This choice was made for practical reasons (we did not release particles deeper than 1000 m) but is justified by the fact that mean sequestration times at this depth are similar to expected times for the overturning circulation of the ocean (i.e., 1000 – 2000 years) and thus we do not expect sequestration duration to increase substantially with increasing depth (and indeed limited simulations suggested that it did not). With that information, the estimate of regional sequestration was calculated as:

$$1.7 \text{ mmol C m}^{-2} \text{ d}^{-1} \times 1370 \text{ years} \times 170000 \text{ km}^2 \times \left(\frac{365 \text{ days}}{\text{year}}\right) \left(\frac{10^6 \text{ m}^2}{\text{km}^2}\right) \left(\frac{12 \text{ mg C}}{\text{mmol C}}\right) \left(\frac{12 \text{ mg C}}{\text{mmol C}}\right) \left(\frac{1 \text{ Pg}}{10^{18} \text{ mg}}\right) = 1.7 \text{ Pg C}$$

Please note that 170000 km² is the approximate area of the study region defined in Supp Fig. 1 (the exact area, which we actually used in our calculations is 169637 km²),

Figure 4: a note needs to be made on the figure/ in the caption that active transport is 0 at 1000m

because fecal pellet production deeper than 450m is not considered and neither is mortality. I know this is highlighted in the text but for those readers who skim the figures, this is an important caveat.

We have added this caveat to the legend, because we certainly do not want to undersell the role of diel vertical migrators.

Line 156-196: These three paragraphs are currently organised as three distinct paragraphs with a literature review, contextualising findings from the study and how the BCP could change elsewhere in relation to each pathway. This makes it hard to follow as a reader with the text constantly jumping between the three pathways. I would suggest integrating the information for each pathway into a paragraph to improve the flow for the reader.

We prefer not to split this into different paragraphs for each pathway, because an important part of this section is to note the interconnected nature of the BCP pathways that we have identified. As one of the other reviewers noted, the pathways are not independent. Carbon that is transported out of the mixed layer by subduction can be consumed by a zooplankter that then migrates deeper and produces a fecal pellet that subsequently sinks even deeper. Our approach artificially separates these related processes and in this section we attempt to make it clear that these processes are actually interconnected.

Line 316: reference 46 has some repetition and doesn't appear to have a title.

Thank you for catching this issue. We have corrected it.

Line 335: Are the effects of different modes of the El Nino Southern Oscillation considerable at the CCE site? Was the sampling equally weighted for different El Nino modes?

Yes, ENSO is the primary driver of interannual variability in the CCE, particularly during spring. The cruises on which active transport and subduction measurements were made were all from ENSO neutral periods. The cruises with sediment traps for sinking particles span different ENSO modes and our approach for integrating these measurements weighted for different ENSO modes. We now make it easy for readers to evaluate our sampling with respect to ENSO modes by including the multi-variate ENSO index in Fig. 1c.

Line 399-410: I appreciate that this is a huge amount of work and that it is necessary to make lots of assumptions i.e. abundance to biomass, specific respiration rate, DOC excretion. I think it is essential to highlight these assumptions briefly in the main body of the text and how this might impact the active transport fluxes/ sequestration budgets. The addition of the active transport pathway is the dominant advance from previous studies (e.g. Stukel et al. 2018) and so I think it requires more discussion on the caveats for those who might not delve into the methodology section.

We believe that we have highlighted the most important of these assumptions in the main text. Specifically, we note that we did not actually make respiration and excretion measurements but rather inferred them from bioenergetics models and allometric relationships and we note that our measurements are only of respiration and excretion and hence do not include other processes like mortality at depth. The specific text from the main text of the manuscript is:

“The biomasses and daytime residence depths of diel vertically-migrating (DVM) zooplankton taxa (copepods, krill, chaetognaths, and hyperiid amphipods) were determined from day-night-paired, vertically-stratified tows from the surface to 450 m³³. DVM mesopelagic fish biomass was quantified using Matsuda-Oozeki-Hu net trawls³⁵. Bioenergetics models and allometric relationships were then used to quantify respiration and excretion of these taxa at their daytime residence depths.”

AND

“Our estimates are, however, likely underestimates of the total magnitude of active transport in the region, because they do not include migrant mortality at depth, which may be responsible for approximately half of total active transport in the region³⁹. Mortality at depth is also likely to lead to deeper remineralization due to the sinking of carcasses and/or fecal pellets of predators that feed on vertical migrants⁴⁰. However, mortality rates in the mesopelagic are highly uncertain due to a paucity of direct measurements and hence we cannot robustly include this process in our carbon budget.”

We believe that this is sufficient in the main text and that repeating much of the methods section in the main text as well would needlessly distract from the main takeaway of the manuscript.

Line 428: What are foraging sorties? The reference doesn't include the word sorties as far as I could tell. Is this a typo?

The Leising manuscript refers to “foraging forays”, which we refer to as “foraging sorties” because this is how it has been referred to in the literature more recently (e.g., Karakoylu, E. M.: The foraging sorties hypothesis : evaluating the effect of gut dynamics on copepod foraging behavior, Scripps Institution of Oceanography, University of California, San Diego, La Jolla, CA, 2010.). To avoid confusion and be consistent with Leising we have replaced sorties with forays.

Line 434: Add model resolution.

We have added that the model has 9-km resolution.

Line 437: Where do the remineralisation rates come from? Apologies if I've missed it but I don't recall this being mentioned.

As outlined in Stukel et al. (2018), remineralization rates (and sinking speed distributions) were determined for each Lagrangian experiment using in situ measurements. More specifically, particle creation rate, particle sinking speed, and particle remineralization rate together constrain particle standing stock in the water column and sinking particle flux. Since we quantified particle creation rate (as net primary production), sinking particle flux (with sediment traps), and particle standing stock we were able to invert the equations to estimate sinking speeds and remineralization rates. Extensive additional details (including specific equations used) are now include in the supplementary methods section.

Line 442-444: What are the particle model and the physical circulation model timesteps? Are the models run so that, for example, the circulation model progresses (let's say with a 1 day timestep) and then the particle model is run within the 1 day (let's say with a 1 hour timestep)? If a particle is moved across the boundary within the 1st hour of the 1 day timestep how do you attribute it to either sinking or subduction? More detail in the text is required here.

The circulation model was run at high temporal resolution but saved with 8-hourly temporal resolution (even at this reduced temporal resolution, the output file for a single 30-day simulation was 2.7GB). These 8-hourly temporal resolution files were then interpolated to 1-hourly resolution which was the time step of the LTRANS Lagrangian model. We considered model particles to have been subducted across a particular depth threshold if they crossed that depth horizon by advection (i.e., during the physical advection subroutine of LTRANS) and to have sunk (gravitationally) if they crossed that depth horizon by sinking (i.e., during the biological subroutine of LTRANS). We have added a supplementary section that explains these details.

Line 448: Am I correct that all active transport and subduction estimates were from El Nino-neutral conditions but that it is not true for the sinking flux estimates? Some comment on how this may impact the averages and contributions from the 3 pathways could be included.

The cruises on which active transport and subduction measurements were made were all from ENSO neutral periods. The cruises with sediment traps for sinking particles span different ENSO modes. Because our approach for averaging the sinking particle results included developing a relationships between export flux and NPP and then applying this result to remotely sensed NPP over a 19 year period (spanning ENSO variability) It directly accounts for ENSO variability . We now make it easy for readers to evaluate our sampling with respect to ENSO modes by including the multi-variate ENSO index in Fig. 1c.

Line 463: I'm not sure about the phrasing of 'remineralized carbon dioxide' – remineralized carbon export or dissolved inorganic carbon would be more precise. Also, see studies that have taken a similar approach and perhaps should be referenced Robinson et al. (2014) and Baker et al. (2022).
<https://agupubs.onlinelibrary.wiley.com/doi/full/10.1002/2013GL058799>
<https://agupubs.onlinelibrary.wiley.com/doi/pdf/10.1029/2021GB007286>

We have changed to “carbon export”. Thank you for pointing us to these studies. We have now cited them in the “An interconnected BCP in a changing ocean” section.

Line 490: It is great that the observational data is available (first and last hyperlink in the file also includes the), which breaks the link and the links are the same). It isn't clear what data is available or how to find it in the edirepository. I searched CCE LTER with no results. Linking to the exact datasets would be much more useful for the reader. The naval research laboratory link is the same as the zooscan database which I assume is a typo. In the future, the authors might consider making the model output available (e.g. zenodo is a free archive) or providing scripts to allow others to replicate the various complex modelling stages that have come together to produce the BCP pathway budgets. As it stands I think it would be rather difficult to replicate this aspect of the work.

Thank you for noting these issues. We have corrected the issues with the CCE DataZoo link and the Naval Research Laboratory. We are surprised that the reviewer did not find any datasets when searching for “CCE LTER”, because we were able to find all of our datasets when using this search term. We do note that the EDI Repository can be difficult to use, however, and that there are multiple different places on the website that offer search options, not all of which actually search the dataset. Consequently, we have provided doi's for our datasets in EDI. Please note that we provide doi's rather than hyperlinks because these should have greater long-term stability. To further ensure replicability, ease of data access, and long-term data availability we also include our processed data as Supp. Tables 1 – 6.

Reviewer #3 (Remarks to the Author):

This manuscript details a large dataset collected over multiple field seasons that characterises the flux of organic carbon by 3 BCP pathways – gravitational sinking, active flux and physical pumps. The analysis seems fairly robust and provides useful conclusions regarding the relative contribution of the different pathways to total flux. However, the authors need to be clearer that these results are only valid for this particular upwelling-dominated region and can't necessarily be viewed as representative of the situation in other parts of the global ocean.

We wholeheartedly agree with this comment and certainly do not mean to imply that our results for this region are representative of global averages. We would argue that our results shed light on the processes that are likely to dominate in other eastern boundary current upwelling biomes and (when specifically investigating spatial variability throughout our region, something that we have highlighted more in the current draft) they shed some light on processes that are important in the North Pacific subtropical gyre. However, we agree that it would be inappropriate to extrapolate our results to, for instance, the Equatorial Pacific, the North Atlantic, or the Southern Ocean. Indeed, we point out that processes are likely quite different in other regions. We have added the section “Spatiotemporal variability in BCP pathways in the CCE” that further addresses some of these issues. We also note that we had made this point clear in the previous (and current) draft of the manuscript:

“We should anticipate, however, that BCP changes will differ substantially in other biomes. For instance, oligotrophic subtropical gyres may be even more heavily weighted towards sinking particles than the CCE, although vertical migrators in oligotrophic regions are likely to reside at even deeper daytime depths^{13,38}. In areas with deep winter mixed layers and/or substantially weaker stratification year-round (e.g., the North Atlantic and Southern Ocean) we might expect that subduction/vertical mixing contributes more to export from the euphotic zone while also having a substantially longer remineralization length scale^{16,18}. Polar and subpolar regions also have ontogenetic (seasonal) vertical migrators that contribute to the “lipid pump¹²”. Additionally, sequestration times are spatially variable with the North Pacific exhibiting some of the longest sequestration periods while the North Atlantic and Southern Ocean have substantially shorter time scales⁵⁶⁻⁵⁸.”

General comment:

The cruises cover several years and times of year, but this temporal variability is ignored for this analysis. That’s fine because it’s the best that can be done with the data available at the moment, but I urge the authors to add a paragraph to the discussion on the likely temporal (seasonal) variability in the relative strength of the pathways and the implications of ignoring temporal variability for their results.

We have added a section “Spatiotemporal variability in BCP pathways in the CCE” that addresses exactly this issue.

There are very large uncertainties in some of the calculations, particularly for the active fluxes. The implications of these large uncertainties and the validity of the underlying assumptions are not currently discussed in the manuscript (and should be). There are also aspects of the active flux not included in the calculations (as described on lines 424-430) which I think should be mentioned in the main body of the manuscript.

We are not certain what the reviewer is referring to in terms of the implications of the large uncertainties in active transport. These large uncertainties are clearly presented in the manuscript and their implications are evident as we propagate these uncertainties through all subsequent calculations.

We also point out that the previous version of our manuscript does clearly describe the fact that our active transport measurements do not include all processes that contribute to export: “Our estimates are, however, likely underestimates of the total magnitude of active transport in the region, because they do not include migrant mortality at depth, which may be responsible for approximately half of total active transport in the region³⁹. Mortality at depth is also likely to lead to deeper remineralization due to the sinking of carcasses and/or fecal pellets of predators that feed on vertical migrants⁴⁰. However, mortality rates in the mesopelagic are highly uncertain

due to a paucity of direct measurements and hence we cannot robustly include this process in our carbon budget.”

We have modified the text slightly to ensure that we also mention foraging forays in the results section. However, we do not believe it is necessary to mention each of the other processes within the results (noting that they are, of course, all mentioned in the methods section) because, apart from mortality at depth and foraging forays, there is no reason to believe that the other processes are important in the CCE region. Ontogenetic vertical migration is more common in polar and subpolar regions. Reverse diel vertical migration, while possible and potentially relevant for specific predator-prey relationships, has not been demonstrated to be biogeochemically important in the CCE. Defecation at depth is likely to be much more important in colder waters in which gut throughput time is substantially longer. At typical CCE temperatures, zooplankton gut passage times are expected to be in the range of 20 – 30 minutes. Thus zooplankton are likely to egest the entirety of their gut before completing a typical vertical migration. While we certainly do not intend to imply that these processes should not be considered, we believe that it is appropriate to mention them only in the methods section, thus allowing ourselves to highlight the two likely important un-included processes within the results section.

Complete active flux estimates are only available for 2 (of 11) cruises, and subduction estimates are only available for 3 (of 11) cruises. The authors need to include some discussion of whether these limited measurements are representative of the conditions in the region, and the influence of interannual/seasonal variability on the interpretation of their results as a consequence.

We agree with the reviewer that spatiotemporal variability in ecosystem processes likely shapes variability in the relative magnitudes of the three BCP pathways quantified in our study. We had addressed (albeit briefly) how differences in ecosystem structure between our cruises (and the fact that we did not sample all pathways on all cruises) might impact our results by noting that the mean sinking particle exports (at 100 m depth) on the experiments for which we quantified active transport and subduction were very close to the regional mean sinking particle export at 100 m depth. However, we can see that this is not sufficient information to really assess the impacts of spatiotemporal variability on our results. Hence we have substantially revised the manuscript to address this topic. We have made two major changes. First, we added three new panels to Figure 1 which show: spatial variability in net primary productivity along with the sampling locations for each cruise (a), temporal variability in NPP along with the times of each cruise (b), and temporal variability in the ENSO index along with the times of each cruise (c). Second, we added a new section entitled “Spatiotemporal variability in BCP pathways in the CCE” that addresses these issues in detail. We have also added a supplemental table (Supp. Table 1) that gives contextual data (temperature, nitrate, chl, NPP) for all Lagrangian experiments.

Having written out these general comments, it seems to me that a separate section is required in the manuscript which covers the assumptions, limitations and uncertainties of the calculations made.

As mentioned above we have added a new section that focuses on most of these questions.

Specific comments:

Line 14: not 'all BCP pathways' are quantified simultaneously here either (lipid pump not included, for example)

Different authors categorize the BCP pathways in different ways. The "lipid pump" is part of what we consider "active transport" as it is carbon transport mediated by organisms that are actively migrating between different water depths (i.e., our "active transport" is the sum of Boyd et al. 2019's "mesopelagic-migrant pump" + "seasonal lipid pump"; similarly our "Subduction and vertical mixing" combines the impacts of the pumps referred to in Boyd et al. as "Eddy subduction pump" + "Mixed layer pump" + "Large-scale physical pump". Within our manuscript we prefer to refer to the "lipid pump" as active transport by ontogenetic vertical migrants. It is true that we did not quantify the impact of active transport by ontogenetic vertical migrants, but that is simply because ontogenetic vertical migrants are not generally dominant taxa in the CCE ecosystem.

Line 20: do the authors truly mean 'carbon dioxide' here, or would 'organic C' be more precise?

We have modified "carbon dioxide" to "carbon" to reflect the fact that the carbon may be stored in multiple forms (i.e., much of it is probably as bicarbonate).

Line 48-49: I think it's stretching credibility a little to claim that the region 'spans much of the ecosystem variability found in the global ocean'

The reviewer is correct, of course, that we do not actually span most of the *ecosystem* variability in the global ocean, because ecosystems vary in many different ways. However, the CCE does contain most of the variability in net primary productivity found in the global ocean, and hence we have modified the sentence to reflect this. (Modified sentence reads: "It <the CCE> thus spans much of the productivity variability found in the global ocean". To support this assertion we have provided the following figure (for review only). It shows a heat map of sea surface temperature vs sea surface chlorophyll for the global ocean from the 1/12 degree SEAWIFS monthly climatology. Superimposed on this map, we include the specific sea surface temperatures and sea surface chlorophylls of our Lagrangian experiments (red diamonds), showing that our study locations encompass much of the global variability in sea surface chlorophyll. We also now

include this sea surface temperature and sea surface chlorophyll data (along with other data) in Supp. Table 1.

Line 51: please include a map of the study region and the sampling regime. It would also be helpful to have an indication (supplementary table?) of the dates and year of each of the cruises.

We have added new panels to figure 1, which include a map of the study region including showing all sampling locations within a plot showing climatological NPP. We have also included time-series of NPP and ENSO (dominant source of interannual variability in our region) with the sampling periods superimposed. This gives readers the ability to judge for themselves how representative (or unrepresentative) our sampling times and locations were.

Line 51 & onwards: reference to the Supplementary Methods needs to be made throughout

Our apologies. Nat. Comm. has a methods section within the main text (at the end of the main text), hence our methods are not in a supplementary section. We thus do not believe it is necessary to remind readers to check the methods section to find our methods in the main text.

Line 60: what is the export depth in this study? Is it uniformly 100m?

Yes, it is uniformly 100-m unless otherwise specified (e.g., in Fig. 3a we follow the conventional definition of the Ez-ratio as based on export out of the euphotic zone following Buesseler & Boyd 2009; in Fig. 3b we show export across three different depth horizons).

Line 77-108: is the active flux remineralisation in Fig 2 the sum of DVM organisms' respiration and excretion?

Yes.

Line 138-143: in the calculations of sequestration potential, do the tracked particles have a decay rate (respiration rate) as they travel? Or is the assumption that the POC remains within the water mass it entered on export until that water mass returns to the surface?

No, since we are simulating carbon dioxide molecules here, they do not have a decay rate but rather are assumed to remain within the water mass until that water mass returns to the surface.

Line 147: is the higher sequestration rate for sinking particles due also to a slower remineralisation rate?

We are confused by this question. Line 147 states: "The higher sequestration due to sinking particles was a result of both a higher flux from the surface ocean ($9.0 \text{ mmol C m}^{-2} \text{ d}^{-1}$ at 100 m depth, compared to 2.9 and 3.8 for active transport and subduction, respectively) and a longer mean sequestration time (586 years, compared to 468 and 279)." The higher sequestration rate is thus due to both higher export from the surface and a longer mean sequestration time. These results come directly from the results outlined in the previous three sections of the manuscript (and summarized in Fig. 2).

Line 443-444: what are the time steps for the particle model and physical model?

The circulation model was run at high temporal resolution but saved with 8-hourly temporal resolution (even at this reduced temporal resolution, the output file for a single 30-day simulation was 2.7GB). These 8-hourly temporal resolution files were then interpolated to 1-hourly resolution which was the time step of the LTRANS Lagrangian model. We considered model particles to have been subducted across a particular depth threshold if they crossed that depth horizon by advection (i.e., during the physical advection subroutine of LTRANS) and to have

sunk (gravitationally) if they crossed that depth horizon by sinking (i.e., during the biological subroutine of LTRANS). We have added a supplementary section that explains these details.

REVIEWERS' COMMENTS

Reviewer #2 (Remarks to the Author):

The authors have provided a robust and convincing rebuttal and they have addressed all the reviewer's comments in a more than satisfactory way. I commend the authors on an excellent study that advances our understanding of BCP sequestration. I recommend the manuscript for publication pending one very minor correction (see below).

The mean sequestration duration at 1000m in Line 422 and in figure 4 are different in the tracked changes document. If they should be the same, please amend the value.

Chelsey Baker

Reviewer #3 (Remarks to the Author):

Second Review of Stukel et al. "Carbon sequestration by multiple biological pump pathways in a coastal upwelling biome"

This manuscript details a large dataset collected over multiple field seasons that characterises the flux of organic carbon by 3 BCP pathways – gravitational sinking, active flux and physical pumps. The analysis seems fairly robust and provides useful conclusions regarding the relative contribution of the different pathways to total flux. However, the authors need to be clearer that these results are only valid for this particular upwelling-dominated region and can't necessarily be viewed as representative of the situation in other parts of the global ocean.

General comment:

The cruises cover several years and times of year, but this temporal variability is ignored for this analysis. That's fine because it's the best that can be done with the data available at the moment, but I urge the authors to add a paragraph to the discussion on the likely temporal (seasonal) variability in the relative strength of the pathways and the implications of ignoring temporal variability for their results.

There are very large uncertainties in some of the calculations, particularly for the active fluxes. The implications of these large uncertainties and the validity of the underlying assumptions are not currently discussed in the manuscript (and should be). There are also aspects of the active flux not included in the calculations (as described on lines 424-430) which I think should be mentioned in the main body of the manuscript.

Complete active flux estimates are only available for 2 (of 11) cruises, and subduction estimates are only available for 3 (of 11) cruises. The authors need to include some discussion of whether these limited measurements are representative of the conditions in the region, and the influence of interannual/seasonal variability on the interpretation of their results as a consequence.

Having written out these general comments, it seems to me that a separate section is required in the manuscript which covers the assumptions, limitations and uncertainties of the calculations made.

Specific comments:

Line 14: not 'all BCP pathways' are quantified simultaneously here either (lipid pump not included, for example)

Line 20: do the authors truly mean 'carbon dioxide' here, or would 'organic C' be more precise?

Line 48-49: I think it's stretching credibility a little to claim that the region 'spans much of the ecosystem variability found in the global ocean'

Line 51: please include a map of the study region and the sampling regime. It would also be helpful to

have an indication (supplementary table?) of the dates and year of each of the cruises.

Line 51 & onwards: reference to the Supplementary Methods needs to be made throughout

Line 60: what is the export depth in this study? Is it uniformly 100m?

Line 77-108: is the active flux remineralisation in Fig 2 the sum of DVM organisms' respiration and excretion?

Line 138-143: in the calculations of sequestration potential, do the tracked particles have a decay rate (respiration rate) as they travel? Or is the assumption that the POC remains within the water mass it entered on export until that water mass returns to the surface?

Line 147: is the higher sequestration rate for sinking particles due also to a slower remineralisation rate?

Line 443-444: what are the time steps for the particle model and physical model?

REVIEWERS' COMMENTS

Reviewer #2 (Remarks to the Author):

The authors have provided a robust and convincing rebuttal and they have addressed all the reviewer's comments in a more than satisfactory way. I commend the authors on an excellent study that advances our understanding of BCP sequestration. I recommend the manuscript for publication pending one very minor correction (see below).

The mean sequestration duration at 1000m in Line 422 and in figure 4 are different in the tracked changes document. If they should be the same, please amend the value.

Thank you for noting this mistake. The correct value is 1335 years and we have updated it in the text at line 422.

Chelsey Baker

Reviewer #3 (Remarks to the Author):

Second Review of Stukel et al. "Carbon sequestration by multiple biological pump pathways in a coastal upwelling biome"

Please note that while this states that it is a second review of our manuscript, the comments from the reviewer are in fact identical to the first set of comments from the reviewer (including reference to now inaccurate line numbers). Since we made substantial revisions to the manuscript between versions 1 and 2 to address many of the excellent points raised by the reviewer, we have not made further changes in response to these identical comments. Hence, below our description of changes made in reference to these comments refers to changes we made between the first submission and the second submission (not between the second and third submissions).

This manuscript details a large dataset collected over multiple field seasons that characterises the flux of organic carbon by 3 BCP pathways – gravitational sinking, active flux and physical pumps. The

analysis seems fairly robust and provides useful conclusions regarding the relative contribution of the different pathways to total flux. However, the authors need to be clearer that these results are only valid for this particular upwelling-dominated region and can't necessarily be viewed as representative of the situation in other parts of the global ocean.

We wholeheartedly agree with this comment and certainly do not mean to imply that our results for this region are representative of global averages. We would argue that our results shed light on the processes that are likely to dominate in other eastern boundary current upwelling biomes and (when specifically investigating spatial variability throughout our region, something that we have highlighted more in the current draft) they shed some light on processes that are important in the North Pacific subtropical gyre. However, we agree that it would be inappropriate to extrapolate our results to, for instance, the Equatorial Pacific, the North Atlantic, or the Southern Ocean. Indeed, we point out that processes are likely quite different in other regions. We have added the section "Spatiotemporal variability in BCP pathways in the CCE" that further addresses some of these issues. We also note that we had made this point clear in the previous (and current) draft of the manuscript:

"We should anticipate, however, that BCP changes will differ substantially in other biomes. For instance, oligotrophic subtropical gyres may be even more heavily weighted towards sinking particles than the CCE, although vertical migrators in oligotrophic regions are likely to reside at even deeper daytime depths^{13,38}. In areas with deep winter mixed layers and/or substantially weaker stratification year-round (e.g., the North Atlantic and Southern Ocean) we might expect that subduction/vertical mixing contributes more to export from the euphotic zone while also having a substantially longer remineralization length scale^{16,18}. Polar and subpolar regions also have ontogenetic (seasonal) vertical migrators that contribute to the "lipid pump"¹². Additionally, sequestration times are spatially variable with the North Pacific exhibiting some of the longest sequestration periods while the North Atlantic and Southern Ocean have substantially shorter time scales⁵⁶⁻⁵⁸."

General comment:

The cruises cover several years and times of year, but this temporal variability is ignored for this analysis. That's fine because it's the best that can be done with the data available at the moment, but I urge the authors to add a paragraph to the discussion on the likely temporal (seasonal) variability in the relative strength of the pathways and the implications of ignoring temporal variability for their results.

We have added a section "Spatiotemporal variability in BCP pathways in the CCE" that addresses exactly this issue.

There are very large uncertainties in some of the calculations, particularly for the active fluxes. The

implications of these large uncertainties and the validity of the underlying assumptions are not currently discussed in the manuscript (and should be). There are also aspects of the active flux not included in the calculations (as described on lines 424-430) which I think should be mentioned in the main body of the manuscript.

We are not certain what the reviewer is referring to in terms of the implications of the large uncertainties in active transport. These large uncertainties are clearly presented in the manuscript and their implications are evident as we propagate these uncertainties through all subsequent calculations.

We also point out that the previous version of our manuscript does clearly describe the fact that our active transport measurements do not include all processes that contribute to export: “Our estimates are, however, likely underestimates of the total magnitude of active transport in the region, because they do not include migrant mortality at depth, which may be responsible for approximately half of total active transport in the region³⁹. Mortality at depth is also likely to lead to deeper remineralization due to the sinking of carcasses and/or fecal pellets of predators that feed on vertical migrants⁴⁰. However, mortality rates in the mesopelagic are highly uncertain due to a paucity of direct measurements and hence we cannot robustly include this process in our carbon budget.”

We have modified the text slightly to ensure that we also mention foraging forays in the results section. However, we do not believe it is necessary to mention each of the other processes within the results (noting that they are, of course, all mentioned in the methods section) because, apart from mortality at depth and foraging forays, there is no reason to believe that the other processes are important in the CCE region. Ontogenetic vertical migration is more common in polar and subpolar regions. Reverse diel vertical migration, while possible and potentially relevant for specific predator-prey relationships, has not been demonstrated to be biogeochemically important in the CCE. Defecation at depth is likely to be much more important in colder waters in which gut throughput time is substantially longer. At typical CCE temperatures, zooplankton gut passage times are expected to be in the range of 20 – 30 minutes. Thus zooplankton are likely to egest the entirety of their gut before completing a typical vertical migration. While we certainly do not intend to imply that these processes should not be considered, we believe that it is appropriate to mention them only in the methods section, thus allowing ourselves to highlight the two likely important un-included processes within the results section.

Complete active flux estimates are only available for 2 (of 11) cruises, and subduction estimates are only available for 3 (of 11) cruises. The authors need to include some discussion of whether these limited measurements are representative of the conditions in the region, and the influence of interannual/seasonal variability on the interpretation of their results as a consequence.

We agree with the reviewer that spatiotemporal variability in ecosystem processes likely shapes variability in the relative magnitudes of the three BCP pathways quantified in our study. We had addressed (albeit briefly) how differences in ecosystem structure between our cruises (and the fact that we did not sample all pathways on all cruises) might impact our results by noting that the mean sinking particle exports (at 100 m depth) on the experiments for which we quantified active transport and subduction were very close to the regional mean sinking particle export at 100 m depth. However, we can see that this is not sufficient information to really assess the impacts of spatiotemporal variability on our results. Hence we have substantially revised the manuscript to address this topic. We have made two major changes. First, we added three new panels to Figure 1 which show: spatial variability in net primary productivity along with the sampling locations for each cruise (a), temporal variability in NPP along with the times of each cruise (b), and temporal variability in the ENSO index along with the times of each cruise (c). Second, we added a new section entitled “Spatiotemporal variability in BCP pathways in the CCE” that addresses these issues in detail. We have also added a supplemental table (Supp. Table 1) that gives contextual data (temperature, nitrate, chl, NPP) for all Lagrangian experiments.

Having written out these general comments, it seems to me that a separate section is required in the manuscript which covers the assumptions, limitations and uncertainties of the calculations made.

As mentioned above we have added a new section that focuses on most of these questions.

Specific comments:

Line 14: not ‘all BCP pathways’ are quantified simultaneously here either (lipid pump not included, for example)

Different authors categorize the BCP pathways in different ways. The “lipid pump” is part of what we consider “active transport” as it is carbon transport mediated by organisms that are actively migrating between different water depths (i.e., our “active transport” is the sum of Boyd et al. 2019’s “mesopelagic-migrant pump” + “seasonal lipid pump”; similarly our “Subduction and vertical mixing” combines the impacts of the pumps referred to in Boyd et al. as “Eddy subduction pump” + “Mixed layer pump” + “Large-scale physical pump”. Within our manuscript we prefer to refer to the “lipid pump” as active transport by ontogenetic vertical migrants. It is true that we did not quantify the impact of active transport by ontogenetic vertical migrants, but that is simply because ontogenetic vertical migrants are not generally dominant taxa in the CCE ecosystem.

Line 20: do the authors truly mean ‘carbon dioxide’ here, or would ‘organic C’ be more precise?

We have modified “carbon dioxide” to “carbon” to reflect the fact that the carbon may be stored in multiple forms (i.e., much of it is probably as bicarbonate).

Line 48-49: I think it’s stretching credibility a little to claim that the region ‘spans much of the ecosystem variability found in the global ocean’

The reviewer is correct, of course, that we do not actually span most of the *ecosystem variability* in the global ocean, because ecosystems vary in many different ways. However, the CCE does contain most of the variability in net primary productivity found in the global ocean, and hence we have modified the sentence to reflect this. (Modified sentence reads: “It <the CCE> thus spans much of the productivity variability found in the global ocean”. To support this assertion we have provided the following figure (for review only). It shows a heat map of sea surface temperature vs sea surface chlorophyll for the global ocean from the 1/12 degree SEAWIFS monthly climatology. Superimposed on this map, we include the specific sea surface temperatures and sea surface chlorophylls of our Lagrangian experiments (red diamonds), showing that our study locations encompass much of the global variability in sea surface chlorophyll. We also now include this sea surface temperature and sea surface chlorophyll data (along with other data) in Supp. Table 1.

Line 51: please include a map of the study region and the sampling regime. It would also be helpful to have an indication (supplementary table?) of the dates and year of each of the cruises.

We have added new panels to figure 1, which include a map of the study region including showing all sampling locations within a plot showing climatological NPP. We have also included time-series of NPP and ENSO (dominant source of interannual variability in our region) with the sampling periods superimposed. This gives readers the ability to judge for themselves how representative (or unrepresentative) our sampling times and locations were.

Line 51 & onwards: reference to the Supplementary Methods needs to be made throughout

Our apologies. Nat. Comm. has a methods section within the main text (at the end of the main text), hence our methods are not in a supplementary section. We thus do not believe it is necessary to remind readers to check the methods section to find our methods in the main text.

Line 60: what is the export depth in this study? Is it uniformly 100m?

Yes, it is uniformly 100-m unless otherwise specified (e.g., in Fig. 3a we follow the conventional definition of the Ez-ratio as based on export out of the euphotic zone following Buesseler & Boyd 2009; in Fig. 3b we show export across three different depth horizons).

Line 77-108: is the active flux remineralisation in Fig 2 the sum of DVM organisms' respiration and excretion?

Yes.

Line 138-143: in the calculations of sequestration potential, do the tracked particles have a decay rate (respiration rate) as they travel? Or is the assumption that the POC remains within the water mass it entered on export until that water mass returns to the surface?

No, since we are simulating carbon dioxide molecules here, they do not have a decay rate but rather are assumed to remain within the water mass until that water mass returns to the surface.

Line 147: is the higher sequestration rate for sinking particles due also to a slower remineralisation rate?

We are confused by this question. Line 147 states: "The higher sequestration due to sinking particles was a result of both a higher flux from the surface ocean ($9.0 \text{ mmol C m}^{-2} \text{ d}^{-1}$ at 100 m

depth, compared to 2.9 and 3.8 for active transport and subduction, respectively) and a longer mean sequestration time (586 years, compared to 468 and 279).” The higher sequestration rate is thus due to both higher export from the surface and a longer mean sequestration time. These results come directly from the results outlined in the previous three sections of the manuscript (and summarized in Fig. 2).

Line 443-444: what are the time steps for the particle model and physical model?

The circulation model was run at high temporal resolution but saved with 8-hourly temporal resolution (even at this reduced temporal resolution, the output file for a single 30-day simulation was 2.7GB). These 8-hourly temporal resolution files were then interpolated to 1-hourly resolution which was the time step of the LTRANS Lagrangian model. We considered model particles to have been subducted across a particular depth threshold if they crossed that depth horizon by advection (i.e., during the physical advection subroutine of LTRANS) and to have sunk (gravitationally) if they crossed that depth horizon by sinking (i.e., during the biological subroutine of LTRANS). We have added a supplementary section that explains these details.